# Causal-StoNet: Causal Inference for High-Dimensional Complex Data

**Yaxin Fang**
Department of Statistics
Purdue University
West Lafayette, IN 47907, USA
fang230@purdue.edu

**Faming Liang**
Department of Statistics
Purdue University
West Lafayette, IN 47907, USA
fmliang@purdue.edu

## Abstract

With the advancement of data science, the collection of increasingly complex datasets has become commonplace. In such datasets, the data dimension can be extremely high, and the underlying data generation process can be unknown and highly nonlinear. As a result, the task of making causal inference with high-dimensional complex data has become a fundamental problem in many disciplines, such as medicine, econometrics, and social science. However, the existing methods for causal inference are frequently developed under the assumption that the data dimension is low or that the underlying data generation process is linear or approximately linear. To address these challenges, this paper proposes a novel causal inference approach for dealing with high-dimensional complex data. The proposed approach is based on deep learning techniques, including sparse deep learning theory and stochastic neural networks, that have been developed in recent literature. By using these techniques, the proposed approach can address both the high dimensionality and unknown data generation process in a coherent way. Furthermore, the proposed approach can also be used when missing values are present in the datasets. Extensive numerical studies indicate that the proposed approach outperforms existing ones.

## 1 Introduction

Causal inference is a fundamental problem in many disciplines such as medicine, econometrics and social science. The problem can be formulated under the potential outcomes framework by Rubin (1974). Let $\boldsymbol{X} \in \mathbb{R}^p$ denote a vector of $p$-dimensional confounders. In this paper, we consider only the binary treatment $A \in \{0, 1\}$, but discuss extensions to multiple-level treatments or continuous treatments later. For each subject at each treatment level $a$, we assume there exists a potential outcome $\boldsymbol{Y}(a)$ that can be observed under the actual treatment. We are interested in estimating the average treatment effect (ATE) $\tau = \mathbb{E}[Y(1) - Y(0)]$. It is known that ATE is identifiable if all confounders that influence both treatment and outcome are observed and the *ignorability* and *overlapping* conditions (see Assumption A1) are satisfied. To estimate ATE, a variety of methods, such as outcome regression, augmented/inverse probability weighting (AIPW/IPW) and matching, have been developed. See Imbens (2004) and Rosenbaum (2002) for overviews. These methods often work under the assumptions:

**Assumption 1.** *(Outcome model) The parametric model $\mu_a(\boldsymbol{X}, \boldsymbol{\theta}_a)$ is a correct specification for the outcome function $\mu_a(\boldsymbol{X}) = \mathbb{E}[\boldsymbol{Y}(a)|\boldsymbol{X}]$, $a \in \{0, 1\}$; i.e., $\mu_a(\boldsymbol{X}) = \mu_a(\boldsymbol{X}, \boldsymbol{\theta}_a^*)$ and $\boldsymbol{\theta}_a^*$ is the true model parameter.*

**Assumption 2.** *(Treatment model) The parametric model $p(\boldsymbol{X}, \boldsymbol{\theta}_s)$ is a correct specification for the propensity score $p(\boldsymbol{X}) = P(A = 1|\boldsymbol{X})$; i.e., $p(\boldsymbol{X}) = p(\boldsymbol{X}, \boldsymbol{\theta}_s^*)$ and $\boldsymbol{\theta}_s^*$ is the true model parameter.*

Let $\hat{\boldsymbol{\theta}}_a$ and $\hat{\boldsymbol{\theta}}_s$ be consistent estimators of $\boldsymbol{\theta}_a^*$ and $\boldsymbol{\theta}_s^*$, respectively. For example, the AIPW estimator of the ATE is given by

$$\hat{\tau}_n = \frac{1}{n} \sum_{i=1}^{n} \left[ \frac{A_i \boldsymbol{Y}_i}{p(\boldsymbol{X}_i, \hat{\boldsymbol{\theta}}_s)} - \frac{A_i - p(\boldsymbol{X}_i, \hat{\boldsymbol{\theta}}_s)}{p(\boldsymbol{X}_i, \hat{\boldsymbol{\theta}}_s)} \mu_1(\boldsymbol{X}_i, \hat{\boldsymbol{\theta}}_1) - \frac{(1 - A_i)Y_i}{1 - p(\boldsymbol{X}_i, \hat{\boldsymbol{\theta}}_s)} - \frac{A_i - p(\boldsymbol{X}_i, \hat{\boldsymbol{\theta}}_s)}{1 - p(\boldsymbol{X}_i, \hat{\boldsymbol{\theta}}_s)} \mu_0(\boldsymbol{X}_i, \hat{\boldsymbol{\theta}}_0) \right], \quad (1)$$

which is doubly robust (Robins et al. (1994)) in the sense that $\hat{\tau}_n$ is consistent if either Assumption 1 or 2 holds and locally efficient if both assumptions hold.

In practice, estimating the ATE often poses two main difficulties: (i) high dimensionality of covariates, which is common in genomic data (Bühlmann, 2013; Schwab et al., 2020), environmental and healthcare data (Antonelli et al., 2019), and social media data (Sharma et al., 2020); and (ii) unknown functional forms of the outcome and propensity score. Various methods have been proposed to address these challenges but in a separate manner. For example, Lasso (Tibshirani, 1996) and other regularization methods have been used to select relevant covariates for high-dimensional problems under the linear model framework (Belloni et al., 2014; Farrell, 2015). On the other hand, deep neural networks have been employed to estimate the outcome and propensity score functions for low-dimensional data (Shi et al., 2019; Farrell et al., 2021). However, none of these methods address both difficulties in a unified manner. Furthermore, missing values are often present in the datasets, which further complicates the causal inference problem (Yang et al., 2019; Guan & Yang, 2019).

Building on existing works on stochastic neural networks (Liang et al., 2022; Sun & Liang, 2022) and sparse deep learning (Liang et al., 2018b; Sun et al., 2022), we propose a Causal Stochastic Neural Network, which is abbreviated as Causal-StoNet in what follows, for addressing the above difficulties encountered in causal inference for high-dimensional complex data. The merits of Causal-StoNet are three-folds:

1. **A natural forward-modeling framework**. As described in Section 2, the StoNet has been formulated as a composition of multiple simple linear and logistic regressions, providing a natural forward-modeling framework for complex data generation processes. In Causal-StoNet, we replace a hidden neuron at an appropriate hidden layer by a visible treatment variable. With its compositional regression structure, Causal-StoNet easily extends to various causal inference scenarios, e.g. missing covariates, multi-level or continuous treatments, and mediation analysis, as discussed in Section 5.

2. **Universal approximation ability**. We prove that the StoNet possesses a valid approximation to a deep neural network, thereby enabling it to possess the universal approximation ability to the outcome and propensity score functions.

3. **Consistent sparse learning**. By imposing an appropriate sparse penalty/prior on the structure of the StoNet, relevant variables to the outcome and propensity score can be identified along with the training of the Causal-StoNet even under the setting of high-dimensional covariates. As a result, the outcome and propensity score can be properly estimated even when their exact functional forms are unknown.

In summary, the Causal-StoNet has successfully tackled the issues of high-dimensional covariates, unknown functional forms, and missing data in a holistic manner, providing a robust and reliable approach of causal inference for high-dimensional complex data.

**Related Works**  In the literature, there are quite a few works employing deep neural networks for causal inference, see e.g., Shi et al. (2019) and Farrell et al. (2021). However, the consistency of the deep neural network estimator is not established in Shi et al. (2019). This property has been studied in Farrell et al. (2021) but under the low-dimensional scenario essentially. Otherwise, it requires the underlying outcome and propensity score functions to be highly smooth with the smoothness degree even higher than the data dimension $p$. In addition, the methods in Shi et al. (2019) or Farrell et al. (2021) cannot perform covariate selection, and they are hard to apply when the dataset contains missing values. Quite recently, Chen et al. (2024) proposed some neural network-based ATE estimators, where only the propensity score or the outcome function is approximated using a neural network.

There are also numerous semi-parametric casual estimation methods in the literature. Causal trees (Athey & Imbens, 2015; Li et al., 2015) develops data-driven approach to estimate heterogeneous treatment effect for subpopulations. Super learner (van der Laan et al., 2007) utilizes an ensemble of different models to enhance the causal effect estimation. Targeted maximum likelihood estimation (van der Laan & Rubin, 2006) proposes a flexible semi-parametric framework based on targeted regularization. These methods can also be combined with deep learning or other machine learning models, but the flexibility of the StoNet in forward modeling of complex data generation processes leads to the uniqueness of Causal-StoNet. It can function effectively in various data scenarios, as discussed in Section 5.

## 2 A Brief Review of Stochastic Neural Networks

The StoNet can be briefly described as follows. Consider a DNN model with $h$ hidden layers. For the sake of simplicity, we assume that the same activation function $\psi(\cdot)$ is used for all hidden units. By separating the feeding and activation operators of each hidden unit, we can rewrite the DNN model in the following form:

$$
\begin{aligned}
\tilde{\boldsymbol{Y}}_1 &= \boldsymbol{b}_1 + \boldsymbol{w}_1 \boldsymbol{X}, \\
\tilde{\boldsymbol{Y}}_i &= \boldsymbol{b}_i + \boldsymbol{w}_i \Psi(\tilde{\boldsymbol{Y}}_{i-1}), \quad i = 2, 3, \dots, h, \\
\boldsymbol{Y} &= \boldsymbol{b}_{h+1} + \boldsymbol{w}_{h+1} \Psi(\tilde{\boldsymbol{Y}}_h) + \boldsymbol{e}_{h+1},
\end{aligned}
\tag{2}
$$

where $\boldsymbol{e}_{h+1} \sim N(0, \sigma_{h+1}^2 I_{d_{h+1}})$ is Gaussian random error; $\tilde{\boldsymbol{Y}}_i, \boldsymbol{b}_i \in \mathbb{R}^{d_i}$ for $i = 1, 2, \dots, h$; $\boldsymbol{Y}, \boldsymbol{b}_{h+1} \in \mathbb{R}^{d_{h+1}}$; $\Psi(\tilde{\boldsymbol{Y}}_{i-1}) = (\psi(\tilde{Y}_{i-1,1}), \psi(\tilde{Y}_{i-1,2}), \dots, \psi(\tilde{Y}_{i-1,d_{i-1}}))^T$ for $i = 2, 3, \dots, h+1$, and $\tilde{Y}_{i-1,j}$ is the $j$th element of $\tilde{\boldsymbol{Y}}_{i-1}$; $\boldsymbol{w}_i \in \mathbb{R}^{d_i \times d_{i-1}}$ for $i = 1, 2, \dots, h+1$, and $d_0 = p$ denotes the dimension of $\boldsymbol{X}$. For simplicity, we consider only the regression problems in (2). By replacing the third equation in (2) with a logit model, the DNN model can be extended to classification problems.

The StoNet is *a probabilistic deep learning model* and constructed by adding auxiliary noise to $\tilde{\boldsymbol{Y}}_i$'s for $i = 1, 2, \dots, h$ in (2). Mathematically, the StoNet model is given by

$$
\begin{aligned}
\boldsymbol{Y}_1 &= \boldsymbol{b}_1 + \boldsymbol{w}_1 \boldsymbol{X} + \boldsymbol{e}_1, \\
\boldsymbol{Y}_i &= \boldsymbol{b}_i + \boldsymbol{w}_i \Psi(\boldsymbol{Y}_{i-1}) + \boldsymbol{e}_i, \quad i = 2, 3, \dots, h, \\
\boldsymbol{Y} &= \boldsymbol{b}_{h+1} + \boldsymbol{w}_{h+1} \Psi(\boldsymbol{Y}_h) + \boldsymbol{e}_{h+1},
\end{aligned}
\tag{3}
$$

as a composition of many simple regressions, where $\boldsymbol{Y}_1, \boldsymbol{Y}_2, \dots, \boldsymbol{Y}_h$ can be viewed as latent variables. Further, we assume that $\boldsymbol{e}_i \sim N(0, \sigma_i^2 I_{d_i})$ for $i = 1, 2, \dots, h+1$. For classification problems, $\sigma_{h+1}^2$ plays the role of temperature for the binomial or multinomial distribution formed at the output layer, and it works with $\{\sigma_1^2, \dots, \sigma_h^2\}$ together to control the variation of the latent variables $\{\boldsymbol{Y}_1, \dots, \boldsymbol{Y}_h\}$.

It has been shown in Liang et al. (2022) that the StoNet is a valid approximator to the DNN, i.e., asymptotically they have the same loss function as the training sample size $n$ becomes large. Let $\boldsymbol{\theta}_i = (\boldsymbol{w}_i, \boldsymbol{b}_i)$, let $\boldsymbol{\theta} = (\boldsymbol{\theta}_1, \boldsymbol{\theta}_2, \cdots, \boldsymbol{\theta}_{h+1})$ denote the parameter vector of the StoNet, let $d_{\boldsymbol{\theta}}$ denote the dimension of $\boldsymbol{\theta}$, and let $\Theta$ denote the space of $\boldsymbol{\theta}$. Let $L : \Theta \to \mathbb{R}$ denote the loss function of the DNN as defined in (2), which is given by

$$
L(\boldsymbol{\theta}) = -\frac{1}{n} \sum_{i=1}^{n} \log \pi(\boldsymbol{Y}^{(i)} | \boldsymbol{X}^{(i)}, \boldsymbol{\theta}),
\tag{4}
$$

where $i$ indexes the training samples. Under appropriate settings for $\sigma_i$'s, the activation function $\psi$, and the parameter space $\Theta$, see Assumption A2 (in Appendix), Liang et al. (2022) showed that the StoNet and the DNN have asymptotically the same training loss function, i.e.,

$$
\sup_{\boldsymbol{\theta} \in \Theta} \left| \frac{1}{n} \sum_{i=1}^{n} \log \pi(\boldsymbol{Y}^{(i)}, \boldsymbol{Y}_{mis}^{(i)} | \boldsymbol{X}^{(i)}, \boldsymbol{\theta}) - \frac{1}{n} \sum_{i=1}^{n} \log \pi(\boldsymbol{Y}^{(i)} | \boldsymbol{X}^{(i)}, \boldsymbol{\theta}) \right| \xrightarrow{p} 0, \quad as \quad n \to \infty,
\tag{5}
$$

where $\boldsymbol{Y}_{mis} = (\boldsymbol{Y}_1, \boldsymbol{Y}_2, \dots, \boldsymbol{Y}_h)$ denotes the collection of all latent variables as defined in (3). The StoNet can work with a wide range of Lipschitz continuous activation functions

such as *tanh*, *sigmoid* and *ReLU*. As explained in Liang et al. (2022), Assumption A2 also restricts the size of the noise added to each hidden unit by setting: $\sigma_1 \le \sigma_2 \le \cdots \le \sigma_{h+1}$, $\sigma_{h+1} = O(1)$, and $d_{h+1}(\prod_{i=k+1}^{h} d_i^2)d_k\sigma_k^2 \prec \frac{1}{h}$ for any $k \in \{1, 2, \ldots, h\}$, where the factor $d_{h+1}(\prod_{i=k+1}^{h} d_i^2)d_k$ can be understood as the amplification factor of the noise $\boldsymbol{e}_k$ at the output layer. In general, the noise added to the first few hidden layers should be small to prevent large random errors propagated to the output layer.

Further, it is assumed that each $\boldsymbol{\theta}$ for the DNN is unique up to some loss-invariant transformations, such as reordering some hidden units and simultaneously changing the signs of some weights and biases, see Liang et al. (2018b) and Sun et al. (2022) for similar assumptions used in the study. Then, under some regularity assumptions for the population negative energy function $Q^*(\boldsymbol{\theta}) = \mathbb{E}(\log \pi(\boldsymbol{Y}|\boldsymbol{X}, \boldsymbol{\theta}))$, see Assumption A3, Liang et al. (2022) showed

$$\|\hat{\boldsymbol{\theta}}_n - \boldsymbol{\theta}^*\| \overset{p}{\to} 0, \quad \text{as } n \to \infty, \tag{6}$$

where $\hat{\boldsymbol{\theta}}_n = \arg\max_{\boldsymbol{\theta}\in\Theta}\{\frac{1}{n}\sum_{i=1}^{n}\log\pi(\boldsymbol{Y}^{(i)}, \boldsymbol{Y}_{mis}^{(i)}|\boldsymbol{X}^{(i)}, \boldsymbol{\theta})\}$, and $\theta^* = \arg\max_{\boldsymbol{\theta}\in\Theta} Q^*(\boldsymbol{\theta})$. That is, the DNN (2) can be trained by training the StoNet (3); they are asymptotically equivalent as $n \to \infty$, thereby the universal approximation property also holds for the StoNet. It is worth noting that in forward prediction, the StoNet ignores auxiliary noise added to hidden neurons and thus performs as the DNN.

## 3 CAUSAL-STONET

### 3.1 THE STRUCTURE

The StoNet provides a unified solution for the challenges faced in causal inference for high-dimensional complex data. In this section, we address the challenges of high-dimensional covariates and unknown functional forms of the outcome and propensity score, leaving the treatment of missing data to Section 5.

Figure 1 illustrates the structure of the Causal-StoNet, where the treatment variable is encompassed as a visible unit in an intermediate hidden layer. The compositional regression architecture of the StoNet ensures seamless computational handling of this setup without introducing any computational complexities. Let $A$ denote the treatment variable. The Causal-StoNet is to learn a decomposition of the joint distribution

$$\pi(\boldsymbol{Y}, \boldsymbol{Y}_{mis}, A|\boldsymbol{X}, \boldsymbol{\theta}) \propto \pi(\boldsymbol{Y}_1|\boldsymbol{X}, \boldsymbol{\theta}_1)\pi(\boldsymbol{Y}_2|\boldsymbol{Y}_1, \boldsymbol{\theta}_2)\pi(A|\boldsymbol{Y}_1, \boldsymbol{\theta}_2)\pi(\boldsymbol{Y}_3|\boldsymbol{Y}_2, A, \boldsymbol{\theta}_3)\pi(\boldsymbol{Y}|\boldsymbol{Y}_3, \boldsymbol{\theta}_4), \tag{7}$$

where $\boldsymbol{Y}_{mis} = (\boldsymbol{Y}_1, \boldsymbol{Y}_2, \boldsymbol{Y}_3)$, $\boldsymbol{\theta} = (\boldsymbol{\theta}_1, \boldsymbol{\theta}_2, \boldsymbol{\theta}_3, \boldsymbol{\theta}_4)$, and $\pi(A|\boldsymbol{Y}_1, \boldsymbol{\theta}_2)$ corresponds to the propensity score. For the visible binary treatment unit, a *sigmoid* activation function can be used for its probability interpretation.

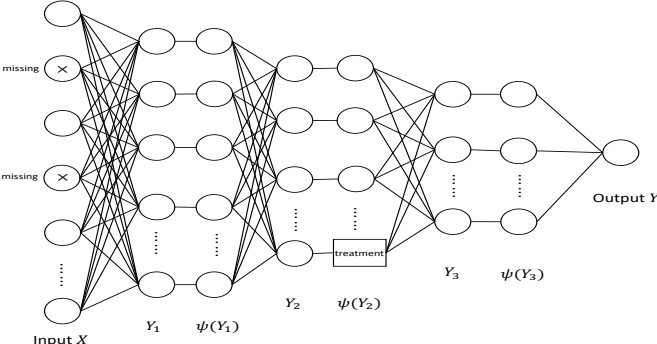

Figure 1: Causal-StoNet Structure: the treatment is included as a visible unit (rectangle) in a middle layer, and $\boldsymbol{Y}_2$ *denotes the latent variable of that layer but with the unit directly feeding to the treatment rectangle excluded;* 'x' represents possible missing values.

To ensure that a sparse Causal-StoNet can be learned for high-dimensional data, where the number of covariates $p$ can be much larger than the sample size $n$, we will follow Sun et al. (2022; 2021) to impose a mixture Gaussian prior on each component of $\boldsymbol{\theta}$, i.e.,

$$\theta \sim \lambda_n N(0, \sigma_{1,n}^2) + (1 - \lambda_n) N(0, \sigma_{0,n}^2), \tag{8}$$

where $\theta$ denotes a generic weight and bias of the Causal-StoNet, $\lambda_n \in (0, 1)$ is the mixture proportion, $\sigma_{0,n}^2$ is typically set to a very small number, while $\sigma_{1,n}^2$ is relatively large.

Let $\mu^*(\boldsymbol{x}, A)$ denote the true outcome function, and let $p^*(\boldsymbol{x})$ denote the true propensity score function. Let $\hat{\mu}(\boldsymbol{x}, A; \hat{\boldsymbol{\theta}}_n)$ denote the DNN estimator of $\mu^*(\boldsymbol{x}, A)$, and let $\hat{p}(\boldsymbol{x}; \hat{\boldsymbol{\theta}}_n)$ denote the DNN estimator of $p^*(\boldsymbol{x})$. For a given estimator $\hat{\boldsymbol{\theta}}_n$, both $\hat{\mu}(\boldsymbol{x}, A; \hat{\boldsymbol{\theta}}_n)$ and $\hat{p}(\boldsymbol{x}; \hat{\boldsymbol{\theta}}_n)$ are calculated as for the conventional DNN model (2) by ignoring the random errors $\boldsymbol{e}_i$'s. Let $\boldsymbol{\gamma}^* = \{\gamma_i : i = 1, 2, \ldots, d_{\boldsymbol{\theta}}\}$ denote the true sparse structure of the Causal-StoNet, which is defined through a sparse DNN as in (A5). Here $\gamma_i$ is an indicator for the existence of connection $c_i$. Following Sun et al. (2022), for each $i \in \{1, 2, \ldots, d_{\boldsymbol{\theta}}\}$, we set $\hat{\gamma}_i = 1$ if the corresponding weight $|\hat{\theta}_i| > \frac{\sqrt{2}\sigma_{0,n}\sigma_{1,n}}{\sqrt{\sigma_{1,n}^2 - \sigma_{0,n}^2}}\sqrt{\log\left(\frac{1-\lambda_n}{\lambda_n}\frac{\sigma_{1,n}}{\sigma_{0,n}}\right)}$ and 0 otherwise. Denote the estimated sparse Causal-StoNet structure by $\hat{\boldsymbol{\gamma}}(\hat{\boldsymbol{\theta}}_n) = \{\hat{\gamma}_i : i = 1, 2, \ldots, d_{\boldsymbol{\theta}}\}$. Under appropriate conditions, we can show that the sparse Causal-StoNet leads to consistent estimates for $\mu^*(\boldsymbol{x}, A)$, $p^*(\boldsymbol{x})$, and $\boldsymbol{\gamma}^*$. This can be summarized as the following theorem, whose proof is given in the appendix.

**Theorem 1.** *Assume that the mixture Gaussian prior (8) is imposed on each connection of the StoNet, Assumptions A2–A5 hold, and $r_n \prec n^{3/16}$. As $n \to \infty$, the following results hold:*

    *(a) (Propensity score function) With probability greater than $1 - \exp\{cn\epsilon_n^2\}$ for some constant $c$,*

$$\mathbb{E}_{\boldsymbol{x}}[(\hat{p}(\boldsymbol{x}; \hat{\boldsymbol{\theta}}_n) - p^*(\boldsymbol{x}))^2] = O\left(\epsilon_n^2 + e^{-cn\epsilon_n^2/16}\right) + o(n^{-1/2}).$$

    *(b) (Outcome function) If $\mu^*(\boldsymbol{x})$ is bounded and the activation function $\psi(\cdot) \in [-1, 1]$, then, with probability greater than $1 - \exp\{cn\epsilon_n^2\}$ for some constant $c$,*

$$\mathbb{E}_{\boldsymbol{x}}(|\hat{\mu}(\boldsymbol{x}, A; \hat{\boldsymbol{\theta}}_n) - \mu^*(\boldsymbol{x})|^2) = O\left((\epsilon_n^2 + e^{-cn\epsilon_n^2})\bar{L}_n^2\right) + o(n^{-1/2}).$$

    *(c) (Structure selection) If Assumption A6 also holds, then $P(\hat{\boldsymbol{\gamma}}(\hat{\boldsymbol{\theta}}_n) = \boldsymbol{\gamma}_*) \xrightarrow{p} 1$.*

By Theorem 1, the sparse Causal-StoNet provides consistent estimators for both the propensity score and outcome functions. Therefore, these estimators can be plugged into equation (1) to get a double robust estimator for ATE. Moreover, the sparse Causal-StoNet also provides consistent identification for the covariates relevant to the treatment and outcome variables, which ensures that the covariates selected for the propensity score function are contained in those selected for the outcome function.

Regarding theoretical properties of the ATE estimator, we have Theorem 2 by following the theory established in Farrell (2015).

**Theorem 2.** *Suppose Assumptions A1–A5 hold. Additionally, assume that the mixture Gaussian prior (8) is imposed on each connection of the StoNet, $r_n \prec n^{3/16}$, and $n^{-1+\xi} \prec \varpi_n^2 \prec n^{-\frac{1}{2}-\xi}$, and specify the network structure such that $0.5 + \xi < \varepsilon < 1 - \xi$ and $\bar{L}_n = O(n^\xi)$ for some $0 \le \xi < 1/4$. Then the following results hold:*

    *(a) $V_\tau^{-1/2}\sqrt{n}(\hat{\tau}_n - \tau^*) \xrightarrow{d} \mathcal{N}(0, 1)$, where $V_\tau$ is given in Supplement A.5, and $\tau^*$ denotes the true value of the ATE.*

    *(b) $\hat{V}_\tau - V_\tau \xrightarrow{p} 1$, where the estimator $\hat{V}_\tau$ is given in Supplement A.5.*

    *(c) (Uniformaly valid inference) Let $\mathcal{P}_n$ be a set of data-generating process satisfying Assumption A1. Then for $c_\alpha = \Phi^{-1}(1 - \alpha/2)$, we have*

$$\sup_{P_n \in \mathcal{P}_n}\left|\mathbb{P}_{P_n}\left[\tau^* \in \left\{\hat{\tau}_n \pm c_\alpha\sqrt{\hat{V}_\tau/n}\right\}\right] - (1 - \alpha)\right| \to 0.$$

We note that by the theory of the StoNet, we can also estimate the propensity score and outcome functions separately by running two sparse DNNs in the way of double machine learning (Chernozhukov et al., 2018). However, in this double machine learning implementation, the covariates selected for the propensity score function might not be a subset of those selected for the outcome function, leading to ambiguity in interpretation for the role that certain covariates play in the causal system. The Causal-StoNet avoids this issue by jointly estimating the propensity score and outcome functions.

## 3.2 Adaptive Stochastic Gradient MCMC for Training Causal-StoNet

As implied by Theorem 1, training the Causal-Stonet can be boiled down to solving a high-dimensional parameter estimation problem with latent variables present, i.e., maximizing

$$\sum_{i=1}^{n} \log \pi(\boldsymbol{Y}^{(i)}, \boldsymbol{Y}_{mis}^{(i)}, A | \boldsymbol{X}^{(i)}, \boldsymbol{\theta}) + \log \pi(\boldsymbol{\theta}). \tag{9}$$

To maximize (9), a feasible method is adaptive stochastic gradient Markov chain Monte Carlo (SGMCMC), which, by the Bayesian version of Fisher's identity (Song et al., 2020), converts the optimization problem to a mean-field equation solving problem:

$$h(\boldsymbol{\theta}) := \int H(\boldsymbol{Y}_{mis}, \boldsymbol{\theta}) d\pi(\boldsymbol{Y}_{mis} | \boldsymbol{X}, \boldsymbol{Y}, A, \boldsymbol{\theta}) = 0, \tag{10}$$

where $H(\boldsymbol{Y}_{mis}, \boldsymbol{\theta}) = \nabla_{\boldsymbol{\theta}} \log \pi(\boldsymbol{Y}, \boldsymbol{Y}_{mis}, A | \boldsymbol{X}, \boldsymbol{\theta}) + \nabla_{\boldsymbol{\theta}} \log \pi(\boldsymbol{\theta})$. The adaptive SGMCMC algorithm works under the framework of stochastic approximation MCMC (Benveniste et al., 1990; Liang et al., 2007). It can be briefly described as follows.

For simplicity of notation, we rewrite (10) in the following equation:

$$h(\boldsymbol{\theta}) = \mathbb{E}[H(\boldsymbol{Z}, \boldsymbol{\theta})] = \int H(\boldsymbol{z}, \boldsymbol{\theta}) \pi(\boldsymbol{z} | \boldsymbol{\theta}) d\boldsymbol{z} = 0, \tag{11}$$

where $\boldsymbol{Z}$ is a latent variable and $\pi(\boldsymbol{z} | \boldsymbol{\theta})$ is a probability density function parameterized by $\boldsymbol{\theta} \in \Theta$. The algorithm works by iterating between the following two steps:

(a) (Sampling) Simulate $\boldsymbol{z}^{(k+1)} \sim \pi(\boldsymbol{z} | \boldsymbol{\theta}^{(k)})$ via a transition kernel induced by a SGMCMC algorithm such as stochastic gradient Langevin dynamics (Welling & Teh, 2011) and stochastic gradient Hamilton Monte Carlo (SGHMC) (Chen et al., 2014).

(b) (Parameter updating) Set $\boldsymbol{\theta}^{(k+1)} = \boldsymbol{\theta}^{(k)} + \gamma_{k+1} H(\boldsymbol{z}^{(k+1)}, \boldsymbol{\theta}^{(k)})$, where $\gamma_{k+1}$ denotes the step size used in the stochastic approximation procedure.

This algorithm is called *adaptive SGMCMC* as the transition kernel used in step (a) changes along with the working estimate $\boldsymbol{\theta}^{(k)}$. Applying the adaptive SGHMC algorithm to (10) leads to Algorithm 1 (in Appendix A.1), where SGHMC is used for simulation of the latent variables $\boldsymbol{Y}_{mis}$ at each iteration. The convergence of the adaptive SGHMC algorithm has been studied in Liang et al. (2022).

**Lemma 1.** *(Liang et al. (2022)) Suppose Assumptions A8-A13 hold. In Algorithm 1, if we set $\epsilon_{k,i} = C_\epsilon/(c_e + k^\alpha)$ and $\gamma_{k,i} = C_\gamma/(c_g + k^\alpha)$ for some constants $\alpha \in (0,1)$, $C_\epsilon > 0$, $C_\gamma > 0$, $c_e \geq 0$ and $c_g \geq 0$, then there exists an iteration $k_0$ and a constant $\lambda_0 > 0$ such that for any $k > k_0$,*

$$\mathbb{E}(\|\hat{\boldsymbol{\theta}}^{(k)} - \hat{\boldsymbol{\theta}}_n^*\|^2) \leq \lambda_0 \gamma_k, \tag{12}$$

*where $\hat{\boldsymbol{\theta}}_n^*$ denotes a solution to equation (10).*

In Liang et al. (2022), an explicit expression of $\lambda_0$ has been given. For simplicity, we have the expression omitted in this paper. Next, Liang et al. (2022) showed that as $k \to \infty$, the imputed latent variable $\boldsymbol{z}^{(k)}$ converges weakly to the desired posterior distribution $\pi(\boldsymbol{z} | \boldsymbol{\theta}^*)$ in 2-Wasserstein distance. Similarly, we establish Lemma 2 which can be used in statistical inference for the problems with missing data being involved.

**Lemma 2.** *Suppose Assumptions A8-A13 hold. Then for any bounded function $\phi_{\boldsymbol{\theta}^*}(\cdot)$, $\widehat{\mathbb{E}_{\mathcal{K}}\phi_{\hat{\boldsymbol{\theta}}^*}(\boldsymbol{z})} - \int \phi_{\hat{\boldsymbol{\theta}}^*}(\boldsymbol{z}) d\pi(\boldsymbol{z} | \hat{\boldsymbol{\theta}}^*) \xrightarrow{p} 0$ as $\mathcal{K} \to \infty$, where $\widehat{\mathbb{E}_{\mathcal{K}}\phi_{\hat{\boldsymbol{\theta}}^*}(\boldsymbol{z})} = \frac{1}{\mathcal{K}} \sum_{i=1}^{\mathcal{K}} \phi_{\hat{\boldsymbol{\theta}}^{(i)}}(\boldsymbol{z}^{(i)})$ and $\{(\hat{\boldsymbol{\theta}}^{(i)}, \boldsymbol{z}^{(i)}) : i = 1, 2, \ldots, \mathcal{K}\}$ denotes a set of parameter estimates and imputed latent variables that are collected in a run of the SGHMC algorithm.*

# 4 NUMERICAL EXAMPLES

## 4.1 SETUP

*Baselines.* We compare Causal-StoNet with the following baselines: (1) Designed for average treatment effect: double selection estimator (**DSE**)(Belloni et al., 2014), approximate residual balancing estimator (**ARBE**) (Athey et al., 2018), targeted maximum likelihood estimator (**TMLE**) (van der Laan & Rubin, 2006), and deep orthogonal networks for unconfounded treatments (**DONUT**) (Hatt & Feuerriegel, 2021); (2) Designed for heterogeneous treatment effect: **X-learner** (Künzel et al., 2017), **Dragonnet**(Shi et al., 2019), causal multi-task deep ensemble (**CMDE**) (Jiang et al., 2023)), causal effect variational autoencoder (**CEVAE**) (Louizos et al., 2017), generative adversarial networks (**GANITE**) (Yoon et al., 2018), and counterfactual regression net (**CFRNet**) [1].

*Performance metrics.* We consider these metrics: (a) estimation accuracy of ATE, which is measured by the mean absolute error (**MAE**) of the ATE estimates; (ii) estimation accuracy of CATE, which is measured by precision in estimation of heterogeneous effect (**PEHE**); and (iii) covariate selection accuracy for the treatment and outcome models, which is measured by false and negative selection rates (**FSR** and **NSR**) as defined in Section A.7.2.

## 4.2 SIMULATION WITH VARYING SAMPLE SIZE

We ran simulated experiment with covariate dimension $p = 1000$ and training sample size $n = 800, 1600, 2400, 3200, 4000$, respectively. For each scenario, 10 simulated datasets are generated as described in Section A.7.1. Both the outcome and treatment effect functions in this experiment are nonlinear. As DSE and ARBE are formulated under linear assumptions, we didn't include them as baselines to ensure a fair comparison. In each experiment, Algorithm 1 was executed 10 times, and the best model was selected based on BIC, as suggested by Sun et al. (2022). This setting will be default for all the experiments unless otherwise stated.

The results are depicted in Figure 2, demonstrating that Causal-StoNet maintains stable performance even with high-dimensional covariates and small sample sizes. Additionally, we conducted further simulations to investigate covariate selection accuracy and address missing value problems, as detailed in Section A.7.2.

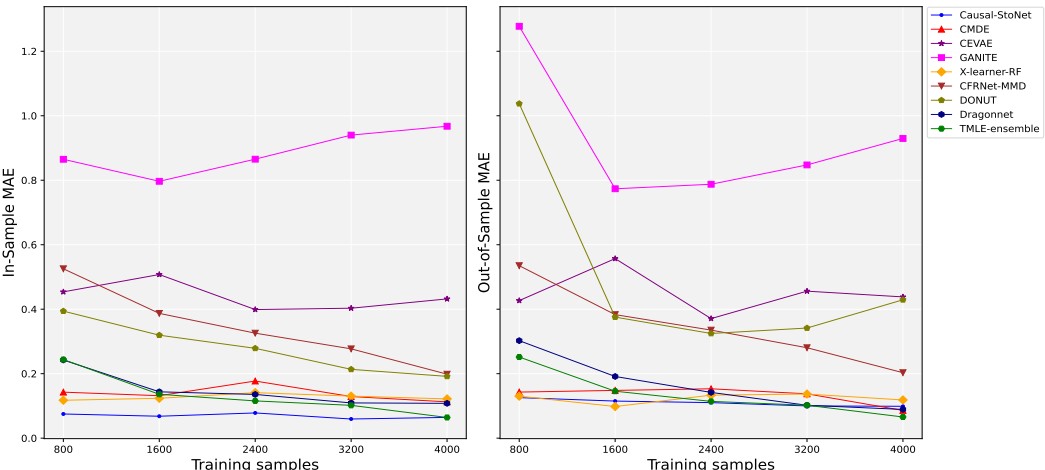

Figure 2: In-sample MAE and Out-of-Sample MAE of ATE estimation with varying training sample sizes. In-sample MAE is calculated over training and validation sets, Out-of-Sample MAE is calculated over test set

---

[1]The code of the experiments is available at: `https://github.com/nixay/Causal-StoNet`

### 4.3 Atlantic Causal Inference Conference 2019 Data Challenge

The Causal-StoNet is compared with baseline methods on 10 synthetic datasets with homogeneous treatment effect from the Atlantic Causal Inference Conference (ACIC) 2019 Data Challenge. Each dataset contains 200 covariates with binary treatment variable, and the outcome variable is continuous. Since they both are synthetic, the true ATE is known. Results in Table 1 demonstrate that Causal-StoNet consistently provides more accurate estimates than the competitive methods.

Table 1: ATE estimation across 10 ACIC 2019 datasets, where the number in the parentheses is the standard deviation of the MAE.

| Method | In-Sample | Out-of-Sample |
|---|---|---|
| Causal-StoNet | **0.0501(0.0118)** | **0.0542(0.0132)** |
| DSE | 0.0776(0.0193) | 0.1632(0.0251) |
| ARBE | 0.0729(0.0166) | 0.1335(0.0179) |
| TMLE(Lasso) | 0.0869(0.0164) | 0.0867(0.0165) |
| TMLE(ensemble) | 0.1140(0.0394) | 0.1316(0.0429) |
| DONUT | 0.5294 (0.2640) | 0.5290(0.2642) |

In addition to ATE, we also consider the conditional average treatment effect (CATE), which measures the heterogeneous treatment effect for subpopulations or individuals based on their covariates $\boldsymbol{x} \in \mathbb{R}^p$ and is defined by

$$\tau(\boldsymbol{x}) = \mathbb{E}[Y(1) - Y(0)|\boldsymbol{X} = \boldsymbol{x}].$$

We evaluated Causal-StoNet's performance in CATE estimation on an ACIC 2019 dataset with heterogeneous treatment effect, where both the treatment and the outcome are binary, using two metrics: $\epsilon_{\text{PEHE}}$ (square root of the Precision in Estimation of Heterogeneous Effect (PEHE)) and $\epsilon_{\text{ATE}}$ (absolute error of estimated ATE). The

Table 2: CATE estimation for an ACIC 2019 dataset.

| Method | $\epsilon_{\text{PEHE}}$ | $\epsilon_{\text{ATE}}$ |
|---|---|---|
| Causal-StoNet | 0.0893 | **0.0118** |
| CMDE | **0.0823** | 0.0444 |
| CMGP | 0.2156 | 0.0258 |
| CEVAE | 0.0867 | 0.0358 |
| GANITE | 0.1913 | 0.0485 |
| X-Learner-RF | 0.1877 | 0.0203 |
| X-Learner-BART | 0.0873 | 0.0720 |
| CFRNet-Wass | 0.1182 | 0.0421 |
| CFRNet-MMD | 0.1158 | 0.0849 |

results in Table 2 show that while Causal-StoNet slightly lags behind CMDE, CEVAE, and X-Learner-BART in $\epsilon_{\text{PEHE}}$, it achieves the lowest $\epsilon_{\text{ATE}}$.

### 4.4 Twins Data

We analyzed a dataset of twin births from 1989 to 1991 in the United States. The treatment variable is binary, with $a = 1$ denoting the heavier twin at birth; and the outcome variable is binary, with $Y = 1$ indicating twin mortality within the first year. We regard each twin-pair's records as potential outcomes, allowing us to find the true ATE. The dataset includes 46 covariates. Refer to A.7.3 for data preprocessing steps. After data pre-processing, we obtained a dataset with 4,821 samples. In this final dataset, mortality rates for lighter and heavier twins are 16.9% and 14.42%, respectively, resulting in a true ATE of $-\mathbf{2.48}\%$. We conducted the experiment in three-fold cross validation, where we partitioned the dataset into three subsets, trained the model using two subsets and estimated the ATE using the remaining one. Table 3 reports the averaged ATE over three folds and the standard deviation of the average. Causal-StoNet yields a more stable ATE estimate than baseline methods.

Table 3: ATE estimates by different methods for twins data

| Causal-StoNet | DSE | ARBE | TMLE(Lasso) | TMLE(ensemble) | DONUT |
|---|---|---|---|---|---|
| **-0.0232(0.0042)** | -0.0405(0.0176) | -0.0096 (0.0201) | -0.1103(0.0599) | -0.1290(0.0779) | -0.0738(0.0128) |

Table A2 shows the covariates selected by Causal-StoNet for the propensity score and outcome models in the three-fold cross-validation experiments. As expected, some covariates that are known to be relevant to the outcome, such as **gestat10**, have been selected for both the treatment and outcome models.

# 5  Some Variants of Causal-StoNet

The proposed Causal-StoNet can be easily extended to various scenarios of causal inference, such as covariates with missing values, multi-level or continuous treatments, and the presence of mediation variables. The extensions can be briefly described as follows.

**Missing at Random (MAR)**  Let $\boldsymbol{X}_{obs}$ denote the observed covariates, let $\boldsymbol{X}_{mis}$ denote the missed covariate values, and let $R$ denote the missing pattern represented as a binary vector. Under the mechanism of missingness at random, i.e., $\boldsymbol{X}_{mis} \perp\!\!\!\perp R | (\boldsymbol{X}_{obs}, A, \boldsymbol{Y})$, the Causal-StoNet as depicted in Figure 1 is to learn a decomposition of the joint distribution

$$
\begin{aligned}
\pi(\boldsymbol{Y}, \boldsymbol{Y}_{mis}, \boldsymbol{X}_{mis}, A | \boldsymbol{X}_{obs}, R, \boldsymbol{\theta}) \propto {} & \pi(\boldsymbol{X}_{mis} | \boldsymbol{X}_{obs}) \pi(\boldsymbol{Y}_1 | \boldsymbol{X}_{obs}, \boldsymbol{X}_{mis}, \theta_1) \pi(\boldsymbol{Y}_2 | \boldsymbol{Y}_1, \theta_2) \\
& \times \pi(A | \boldsymbol{Y}_1, \theta_2) \pi(\boldsymbol{Y}_3 | \boldsymbol{Y}_2, A, \theta_3) \pi(\boldsymbol{Y} | \boldsymbol{Y}_3, \theta_4),
\end{aligned} \tag{13}
$$

where $\boldsymbol{Y}_{mis} = (\boldsymbol{Y}_1, \boldsymbol{Y}_2, \boldsymbol{Y}_3)$, $\boldsymbol{\theta} = (\theta_1, \theta_2, \theta_3, \theta_4)$, $\pi(A | \boldsymbol{Y}_1, \theta_2)$ corresponds to the propensity score, and $\pi(\boldsymbol{X}_{mis} | \boldsymbol{X}_{obs})$ can be formulated in graphical models (see e.g. Liang et al. (2018a)) and will not be detailed here. It is easy to see that in this scenario, the Causal-StoNet can still be trained using Algorithm 1 by treating $\boldsymbol{X}_{mis}$ as part of the latent variables. Statistical inference with imputed missing data can then be made based on Lemma 2.

**Missing not at Random (MNAR)**  The Causal-StoNet can also be extended to the scenario of MNAR, where the missing pattern depends on the missing values themselves even after controlling for observed data. To make the full data distribution identifiable, following Yang et al. (2019), we will assume that the missing pattern $R$ is independent of the outcome given the treatment and confounders, i.e., $\boldsymbol{Y} \perp\!\!\!\perp R | (A, \boldsymbol{X}_{obs}, \boldsymbol{X}_{mis})$. Under this assumption,

$$
\begin{aligned}
\pi(\boldsymbol{Y}, \boldsymbol{Y}_{mis}, \boldsymbol{X}_{mis}, A | \boldsymbol{X}_{obs}, R, \boldsymbol{\theta}) \propto {} & \pi(\boldsymbol{X}_{mis} | \boldsymbol{X}_{obs}) \pi(A | \boldsymbol{X}_{obs}, \boldsymbol{X}_{mis}, \boldsymbol{\theta}) \pi(R | \boldsymbol{X}_{obs}, \boldsymbol{X}_{mis}, A, \boldsymbol{\theta}) \\
& \times \pi(\boldsymbol{Y} | \boldsymbol{X}_{obs}, \boldsymbol{X}_{mis}, A, \boldsymbol{\theta}).
\end{aligned}
$$

To accommodate the term $\pi(R | \boldsymbol{X}_{obs}, \boldsymbol{X}_{mis}, A, \boldsymbol{\theta})$ in the decomposition, we can include some extra visible units for $R$ at some layer between the treatment layer and the output layer. Note that the $R$ units will not be forwardly connected to the output layer.

**Multilevel or Continuous Treatment Variables**  The extension of the Causal-StoNet to this scenario is straightforward. For continuous treatment variable, the Causal-StoNet as depicted in Figure 1 can be directly applied with an appropriate modification of the activation function for the treatment neuron. For multilevel treatment variable, we can simply include multiple visible treatment neurons in the sample hidden layer, with a softmax activation function being used for them.

**Causal Mediation Analysis**  In this scenario, we aim to measure how the treatment effect is affected by intermediate/mediation variables. For example, Pearl (2001) gave an example where the side effect of a drug may cause patients to take aspirin, and the latter has a separate effect on the disease that the drug was originally prescribed for. The mediation analysis can be easily conducted with the Causal-StoNet by including some extra visible units for mediation variables at some layer between the treatment layer and the output layer. The mediation units was fed by the treatment unit and other hidden units of the same layer, and then feeds forward to cast its effect on the outcome layer.

# 6  Conclusion

We have developed an effective method for causal inference with high-dimensional complex data, which addresses the difficulties, including high-dimensional covariates, unknown treatment and outcome functional forms, and missing data, that are frequently encountered in the practice of modern data science. The proposed method does not only possess attractive theoretical properties, but also numerically outperforms the existing methods as demonstrated by our extensive examples.

The Causal-StoNet introduces an innovative deep neural network structure, incorporating visible neurons in its middle layers. Its stochastic deep learning nature renders Causal-StoNet essentially a universal tool for causal inference. It can model complex data generation processes in a forward manner, consistently identify relevant features, and provide accurate approximation to the underlying functions. Furthermore, the flexibility of adaptive SGMCMC algorithms, which impute latent variables (and handle missing data) while consistently estimating model parameters, greatly facilitates the computation of Causal-StoNet.

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

## A  APPENDIX

### A.1  ADAPTIVE STOCHASTIC GRADIENT HAMILTONIAN MONTE CARLO ALGORITHM

Let $(\boldsymbol{Y}_{0,a}^{(s,k)}, \boldsymbol{Y}_{h+1,a}^{(s,k)}) = (\boldsymbol{X}^{(s)}, \boldsymbol{Y}^{(s)}, A^{(s)})$ denote a training sample $s$, and let $\boldsymbol{Y}_{mis,a}^{(s,k)} = (\boldsymbol{Y}_{1,a}^{(s,k)}, \ldots, \boldsymbol{Y}_{h,a}^{(s,k)})$ denote the latent variables imputed for the training sample $s$ at iteration $k$, where the subscript $a$ indicates that the imputed values are affected by the treatment variable $A$.

**Algorithm 1:** An Adaptive SGHMC algorithm for training StoNet

**Input**: total iteration number $K$, Monte Carlo step number $t_{MC}$, the learning rate
sequence $\{\epsilon_{k,i} : k = 1, 2, \ldots, K; i = 1, 2, \ldots, h+1\}$, and the step size sequence
$\{\gamma_{k,i} : k = 1, 2, \ldots, K; i = 1, 2, \ldots, h+1\}$;
and the step size sequence $\{\gamma_{k,i} : k = 1, 2, \ldots, K; i = 1, 2, \ldots, h+1\}$;

**Initialization**: Randomly initialize the network parameters $\hat{\boldsymbol{\theta}}^{(0)} = (\hat{\theta}_1^{(0)}, \ldots, \hat{\theta}_{h+1}^{(0)})$;

**for** $k=1,2,\ldots,K$ **do**

  **STEP 0: Subsampling**: Draw a mini-batch of data and denote it by $S_k$;
  **STEP 1: Backward Sampling**

  For each observation $s \in S_k$, sample $\boldsymbol{Y}_{i,a}^{(s,k)}$'s, in the order from layer $h$ to layer 1,
  from

  $$\pi(\boldsymbol{Y}_{i,a}^{(s,k)}|\hat{\theta}_i^{(k-1)}, \hat{\theta}_{i+1}^{(k-1)}, \boldsymbol{Y}_{i+1,a}^{(s,k)}, \boldsymbol{Y}_{i-1,a}^{(s,k)}) \propto \pi(\boldsymbol{Y}_{i+1,a}^{(s,k)}|\hat{\theta}_{i+1}^{(k-1)}, \boldsymbol{Y}_{i,a}^{(s,k)})\pi(\boldsymbol{Y}_{i,a}^{(s,k)}|\hat{\theta}_i^{(k-1)}, \boldsymbol{Y}_{i-1,a}^{(s,k)}),$$

  by running SGHMC in $k_{MC}$ steps:
  Initialize $\boldsymbol{v}_i^{(s,0)} = \boldsymbol{0}$, and initialize $\boldsymbol{Y}_{i,a}^{(s,k,0)}$ by the corresponding $\tilde{\boldsymbol{Y}}_i$ calculated in (2).
  **for** $l = 1, 2, \ldots, t_{MC}$ **do**
    **for** $i = h, h-1, \ldots, 1$ **do**

$$\boldsymbol{v}_i^{(s,k,l)} = (1 - \epsilon_{k,i}\eta)\boldsymbol{v}_i^{(s,k,l-1)} + \epsilon_{k,i}\nabla_{\boldsymbol{Y}_{i,a}^{(s,k,l-1)}} \log \pi\left(\boldsymbol{Y}_{i,a}^{(s,k,l-1)} \mid \hat{\theta}_i^{(k-1)}, \boldsymbol{Y}_{i-1,a}^{(s,k,l-1)}\right)$$
$$+ \epsilon_{k,i}\nabla_{\boldsymbol{Y}_{i,a}^{(s,k,l-1)}} \log \pi\left(\boldsymbol{Y}_{i+1,a}^{(s,k,l-1)} \mid \hat{\theta}_{i+1}^{(k-1)}, \boldsymbol{Y}_{i,a}^{(s,k,l-1)}\right) + \sqrt{2\epsilon_{k,i}\eta}\boldsymbol{e}^{(s,k,l)},$$
$$\boldsymbol{Y}_{i,a}^{(s,k,l)} = \boldsymbol{Y}_{i,a}^{(s,k,l-1)} + \epsilon_{k,i}\boldsymbol{v}_i^{(s,k,l-1)},$$

(A1)

      where $\boldsymbol{e}^{s,k,l} \sim N(0, \boldsymbol{I}_{d_i})$, $\epsilon_{k,i}$ is the learning rate, and $\eta$ is the friction
      coefficient.
    **end**
  **end**

  Set $\boldsymbol{Y}_{i,a}^{(s,k)} = \boldsymbol{Y}_{i,a}^{(s,k,t_{MC})}$ for $i = 1, 2, \ldots, h$.
  **STEP 2: Parameter Update**
  Update $\hat{\boldsymbol{\theta}}^{(k)} = (\hat{\theta}_1^{(k)}, \hat{\theta}_2^{(k)}, \ldots, \hat{\theta}_{h+1}^{(k)})$ by stochastic approximation Robbins & Monro
  (1951):

$$\hat{\theta}_i^{(k)} = \hat{\theta}_i^{(k-1)} + \gamma_{k,i} \left( \sum_{s \in S_k} \nabla_{\theta_i} \log \pi(Y_{i,a}^{(s,k)}|\hat{\theta}_i^{(k-1)}, Y_{i-1,a}^{(s,k)}) + \frac{|S_k|}{n}\nabla_{\boldsymbol{\theta}_i} \log \pi(\boldsymbol{\theta}) \right),$$

(A2)

  for $i = 1, 2, \ldots, h+1$, where $\gamma_{k,i}$ is the step size used for updating $\theta_i$.
**end**

A.2   ASSUMPTIONS ON DATA GENERATING PROCESS

Suppose that the dataset $\{(\boldsymbol{y}_i, a_i, \boldsymbol{x}_i) : i = 1, 2, \ldots, n\}$ is generated from a process $P_n$, where
$a_i \in \{0, 1\}$, $\boldsymbol{Y}(a) = \mu_a(\boldsymbol{X}) + U$, and $U$ denotes random error. $P_n$ obeys Assumption A1,
with bounds uniformly in $n$:

**Assumption A1.**     *(a) $\{(y_i, a_i, \boldsymbol{x}_i) : i = 1, 2, \ldots, n\}$ is an i.i.d sample from $(\boldsymbol{Y}, A, \boldsymbol{X})$.*

   *(b) (Ignorability) $\{\boldsymbol{Y}(0), \boldsymbol{Y}(1)\} \perp\!\!\!\perp A|\boldsymbol{X}$.*

   *(c) (Overlap) $0 < \zeta_1 \leq P(A = 1|\boldsymbol{X}) \leq \zeta_2 < 1$ almost surely for some $\zeta_1$ and $\zeta_2$.*

   *(d) $\mathbb{E}[|U|^4|\boldsymbol{X}] \leq \mathcal{U}$ for some $\mathcal{U} > 0$.*

   *(e) For some $\delta > 0$, $\mathbb{E}[|\mu_a(\boldsymbol{X})\mu_{1-a}(\boldsymbol{X})|^{1+\delta}]$ and $\mathbb{E}[|U|^{4+\delta}]$ are bounded.*

## A.3 Assumptions for StoNets

The property of the StoNet as an approximator to the DNN, i.e., asymptotically they have the same loss function as the training sample size $n \to \infty$, has been studied in Liang et al. (2022). A brief review for their theory is provided as follows, which form the basis for this work.

Let $\boldsymbol{\theta} = (\boldsymbol{w}_1, \boldsymbol{b}_1, \ldots, \boldsymbol{w}_{h+1}, \boldsymbol{b}_{h+1})$ denote the collection of all weights of the StoNet (3), let $\boldsymbol{\Theta}$ denote the space of $\boldsymbol{\theta}$, let $\boldsymbol{Y}_{\mathrm{mis}} = (\boldsymbol{Y}_1, \boldsymbol{Y}_2, \ldots, \boldsymbol{Y}_h)$ denote the collection of all latent variables, let $\pi(\boldsymbol{Y}, \boldsymbol{Y}_{\mathrm{mis}}|\boldsymbol{X}, \boldsymbol{\theta})$ denote the likelihood function of the StoNet, and let $\pi_{\mathrm{DNN}}(\boldsymbol{Y}|\boldsymbol{X}, \boldsymbol{\theta})$ denote the likelihood function of the DNN model (2). Regarding the network structure, activation function and the variance of the latent variables, they made the following assumption:

**Assumption A2.** *(i) $\boldsymbol{\Theta}$ is compact, i.e., $\boldsymbol{\Theta}$ is contained in a $d_\theta$-ball centered at 0 with radius $r$; (ii) $\mathbb{E}(\log \pi(\boldsymbol{Y}, \boldsymbol{Y}_{\mathrm{mis}}|\boldsymbol{X}, \boldsymbol{\theta}))^2 < \infty$ for any $\boldsymbol{\theta} \in \boldsymbol{\Theta}$; (iii) the activation function $\psi(\cdot)$ is $c'$-Lipschitz continuous for some constant $c'$; (iv) the network's depth $h$ and widths $d_l$'s are both allowed to increase with $n$; (v) $\sigma_1 \leq \sigma_2 \leq \cdots \leq \sigma_{h+1}$, $\sigma_{h+1} = O(1)$, and $d_{h+1}(\prod_{i=k+1}^{h} d_i^2)d_k \sigma_k^2 \prec \frac{1}{h}$ for any $k \in \{1, 2, \ldots, h\}$.*

Assumption A2-(iii) allows the StoNet to work with a wide range of Lipschitz continuous activation functions such as *tanh, sigmoid* and *ReLU*. Assumption A2-(v) constrains the size of noise added to each hidden neuron, where the factor $d_{h+1}(\prod_{i=k+1}^{h} d_i^2)d_k$ can be understood as the amplification factor of the noise $\boldsymbol{e}_k$ at the output layer. In general, the noise added to the first few hidden layers should be small to prevent large random errors propagated to the output layer. Under Assumption A2, they proved the result (5).

Further, regarding the equivalence between training the StoNet and the DNN, they made the following assumption regarding the energy surface of the DNN. Let $Q^*(\boldsymbol{\theta}) = \mathbb{E}(\log \pi_{\mathrm{DNN}}(\boldsymbol{Y}|\boldsymbol{X}, \boldsymbol{\theta}))$, where the expectation is taken with respect to the joint distribution $\pi(\boldsymbol{X}, \boldsymbol{Y})$. By Assumption A2-(i)&(ii) and the law of large numbers,

$$\frac{1}{n} \sum_{i=1}^{n} \log \pi_{\mathrm{DNN}}(\boldsymbol{Y}^{(i)}|\boldsymbol{X}^{(i)}, \boldsymbol{\theta}) - Q^*(\boldsymbol{\theta}) \xrightarrow{p} 0 \tag{A3}$$

holds uniformly over $\boldsymbol{\Theta}$. They assumed $Q^*(\boldsymbol{\theta})$ satisfies the following regularity conditions:

**Assumption A3.** *(i) $Q^*(\boldsymbol{\theta})$ is continuous in $\boldsymbol{\theta}$ and uniquely maximized at $\boldsymbol{\theta}^*$; (ii) for any $\epsilon > 0$, $\sup_{\boldsymbol{\theta} \in \boldsymbol{\Theta} \backslash B(\epsilon)} Q^*(\boldsymbol{\theta})$ exists, where $B(\epsilon) = \{\boldsymbol{\theta} : \|\boldsymbol{\theta} - \boldsymbol{\theta}^*\| < \epsilon\}$, and $\delta = Q^*(\boldsymbol{\theta}^*) - \sup_{\boldsymbol{\theta} \in \boldsymbol{\Theta} \backslash B(\epsilon)} Q^*(\boldsymbol{\theta}) > 0$.*

Assumption A3 restricts the shape of $Q^*(\boldsymbol{\theta})$ around the global maximizer, which cannot be discontinuous or too flat. Given nonidentifiability of the neural network model, Assumption A3 has implicitly assumed that each $\boldsymbol{\theta}$ is unique up to the loss-invariant transformations, e.g., reordering the hidden neurons of the same hidden layer and simultaneously changing the signs of some weights and biases. Under Assumptions A2 and A3, they proved the result (6). In summary, we have the following lemma for the StoNet:

**Lemma A1.** *(Liang et al., 2022) Suppose Assumptions A2 and A3 hold, and $\pi(\boldsymbol{Y}, \boldsymbol{Y}_{\mathrm{mis}}|\boldsymbol{X}, \boldsymbol{\theta})$ is continuous in $\boldsymbol{\theta}$. Then (5) and (6) hold.*

## A.4 Proof of Theorem 1

To prove Theorem 1, we first give a brief review for the theory of sparse deep learning Sun et al. (2022). The theory under the context of Bayesian deep neural networks where each parameter, including the bias and connection weights, is subject to a mixture Gaussian prior distribution.

Assume that the distribution of $y$ given $\boldsymbol{x}$ is given in a generalized linear model as

$$p(y, \boldsymbol{x}; \boldsymbol{\theta}) = \exp\{A(\mu^*(\boldsymbol{x}))y + B(\mu^*(\boldsymbol{x})) + C(y)\},$$

where $\mu^*(\boldsymbol{x})$ denotes a nonlinear function of $\boldsymbol{x}$, and $A(\cdot)$, $B(\cdot)$ and $C(\cdot)$ are appropriately defined functions.

Motivated by the universal approximation ability of the DNN, they proposed to approximate $\mu^*(\boldsymbol{x})$ using a DNN with $H_n - 1$ hidden layers and $L_h$ hidden units at layer $h$, where $L_{H_n} = 1$ for the output layer and $L_0 = p_n$ for the input layer. Let $\boldsymbol{w}^h \in \mathbb{R}^{L_h \times L_{h-1}}$ and $\boldsymbol{b}^h \in \mathbb{R}^{L_h \times 1}$, $h \in \{1, 2, ..., H_n\}$ denote the weights and bias of layer $h$, and let $\psi^h : R^{L_h \times 1} \rightarrow \mathbb{R}^{L_h \times 1}$ denote a coordinate-wise and piecewise differentiable activation function of layer $h$. The DNN forms a nonlinear mapping

$$\mu(\boldsymbol{\theta}, \boldsymbol{x}) = \boldsymbol{w}^{H_n} \psi^{H_n-1} \left[ \cdots \psi^1 \left[ \boldsymbol{w}^1 \boldsymbol{x} + \boldsymbol{b}^1 \right] \cdots \right] + \boldsymbol{b}^{H_n}, \tag{A4}$$

where $\boldsymbol{\theta} = (\boldsymbol{w}, \boldsymbol{b}) = \{w_{ij}^h, b_k^h : h \in \{1, 2, ..., H_n\}, i, k \in \{1, ..., L_h\}, j \in \{1, ..., L_{h-1}\}\}$ denotes the collection of all weights and biases, consisting of $K_n = \sum_{h=1}^{H_n} (L_{h-1} \times L_h + L_h)$ elements in total. To facilitate representation of the sparse DNN, they further introduced an indicator variable for each weight and bias of the DNN, which indicates the existence of the connection in the network. Let $\boldsymbol{\gamma}^{\boldsymbol{w}^h}$ and $\boldsymbol{\gamma}^{\boldsymbol{b}^h}$ denote the matrix and vector of the indicator variables associated with $\boldsymbol{w}^h$ and $\boldsymbol{b}^h$, respectively. Let $\boldsymbol{\gamma} = \{\boldsymbol{\gamma}_{ij}^{\boldsymbol{w}^h}, \boldsymbol{\gamma}_k^{\boldsymbol{b}^h} : h \in \{1, 2, ..., H_n\}, i, k \in \{1, ..., L_h\}, j \in \{1, ..., L_{h-1}\}\}$ and $\boldsymbol{\theta}_{\boldsymbol{\gamma}} = \{w_{ij}^h, b_k^h : \boldsymbol{\gamma}_{ij}^{\boldsymbol{w}^h} = 1, \boldsymbol{\gamma}_k^{\boldsymbol{b}^h} = 1, h \in \{1, 2, ..., H_n\}, i, k \in \{1, ..., L_h\}, j \in \{1, ..., L_{h-1}\}\}$, which specify, respectively, the structure and associated parameters for a sparse DNN.

Among many DNNs that can provide a good approximation to $\mu^*(\boldsymbol{x})$, they define the *true DNN* model as

$$(\boldsymbol{\theta}^*, \boldsymbol{\gamma}^*) = \underset{(\boldsymbol{\theta}, \boldsymbol{\gamma}) \in \mathcal{G}_n, \, \|\mu(\boldsymbol{\theta}, \boldsymbol{\gamma}, \boldsymbol{x}) - \mu^*(\boldsymbol{x})\|_{L^2(\Omega)} \leq \varpi_n}{\arg\min} |\boldsymbol{\gamma}|, \tag{A5}$$

where $\mathcal{G}_n := \mathcal{G}(C_0, C_1, \varepsilon, p_n, H_n, L_1, L_2, \ldots, L_{H_n})$ denotes the space of valid sparse DNNs satisfying condition Assumption A4-(A.2) for the given values of $H_n$, $p_n$, and $L_h$'s, and $\varpi_n$ is some sequence converging to 0 as $n \rightarrow \infty$. For any given DNN $(\boldsymbol{\theta}, \boldsymbol{\gamma})$, the error $\mu(\boldsymbol{\theta}, \boldsymbol{\gamma}, \boldsymbol{x}) - \mu^*(\boldsymbol{x})$ can be generally decomposed as the network approximation error $\mu(\boldsymbol{\theta}^*, \boldsymbol{\gamma}^*, \boldsymbol{x}) - \mu^*(\boldsymbol{x})$ and the network estimation error $\mu(\boldsymbol{\theta}, \boldsymbol{\gamma}, \boldsymbol{x}) - \mu(\boldsymbol{\theta}^*, \boldsymbol{\gamma}^*, \boldsymbol{x})$. They generally treated $\varpi_n$ as the network approximation error. In addition, they made the following two assumptions.

**Assumption A4.** *A.1 The input $\boldsymbol{x}$ is bounded by 1 entry-wisely, i.e. $\boldsymbol{x} \in \mathcal{X} = [-1, 1]^{p_n}$, and the density of $\boldsymbol{x}$ is bounded in its support $\mathcal{X}$ uniformly with respect to $n$.*

*A.2 The true sparse DNN model satisfies the following conditions:*

*A.2.1 The network structure satisfies: $r_n H_n \log n + r_n \log \overline{L}_n + s_n \log p_n \leq C_0 n^{1-\varepsilon}$, where $0 < \varepsilon < 1$ is a small constant, $r_n = |\boldsymbol{\gamma}^*|$ denotes the connectivity of $\boldsymbol{\gamma}^*$, $\overline{L}_n = \max_{1 \leq j \leq H_n-1} L_j$ denotes the maximum hidden layer width, $s_n$ denotes the input dimension of $\boldsymbol{\gamma}^*$.*

*A.2.2 The network weights are polynomially bounded: $\|\boldsymbol{\beta}^*\|_\infty \leq E_n$, where $E_n = n^{C_1}$ for some constant $C_1 > 0$.*

*A.3 The activation function $\psi$ is Lipschitz continuous with a Lipschitz constant of 1.*

**Assumption A5.** *The mixture Gaussian prior (8) satisfies the conditions: $\lambda_n = O(1/\{K_n[n^{H_n}(\overline{L}_n p_n)]^\tau\})$ for some constant $\tau > 0$, $E_n/\{H_n \log n + \log \overline{L}_n\}^{1/2} \lesssim \sigma_{1,n} \lesssim n^\alpha$ for some constant $\alpha > 0$, and $\sigma_{0,n} \lesssim \min\{1/\{\sqrt{n}K_n(n^{3/2}\sigma_{1,0}/H_n)^{H_n}\}, 1/\{\sqrt{n}K_n(nE_n/H_n)^{H_n}\}\}$.*

Based on the two assumptions, Sun et al. (2022) proved the following result on posterior consistency of the sparse DNN.

**Lemma A2.** *(Theorem 2.1; Sun et al. (2022)) Suppose Assumptions A4-A5 hold. Then there exists an error sequence $\epsilon_n^2 = O(\varpi_n^2) + O(\zeta_n^2)$ such that $\lim_{n\rightarrow\infty} \epsilon_n = 0$ and $\lim_{n\rightarrow\infty} n\epsilon_n^2 = \infty$, and the posterior distribution satisfies*

$$\mathbb{P}\left\{\pi[d(p_{\boldsymbol{\theta}}, p_{\mu^*}) > 4\epsilon_n | D_n] \geq 2e^{-cn\epsilon_n^2}\right\} \leq 2e^{-cn\epsilon_n^2},$$
$$\mathbb{E}_{D_n}\pi[d(p_{\boldsymbol{\theta}}, p_{\mu^*}) > 4\epsilon_n | D_n] \leq 4e^{-2cn\epsilon_n^2}, \tag{A6}$$

*for sufficiently large $n$, where $c$ denotes a constant, $D_n$ denotes a dataset of $n$ i.i.d. observations, $\zeta_n^2 = [r_n H_n \log n + r_n \log \overline{L}_n + s_n \log p_n]/n$, $p_{\mu^*}$ denotes the underlying true data*

*distribution, and $p_{\boldsymbol{\theta}}$ denotes the data distribution reconstructed by the Bayesian DNN based on its posterior samples.*

It is known that the DNN model is generally nonidentifiable due to the symmetry of the network structure. For example, the approximation $\mu(\boldsymbol{\theta}, \boldsymbol{\gamma}, \boldsymbol{x})$ can be invariant if one permutes the orders of certain hidden nodes, simultaneously changes the signs of certain weights and biases if *tanh* is used as the activation function, or re-scales certain weights and bias if Relu is used as the activation function. To address this issue, Sun et al. (2022) considered a set of DNNs, denoted by $\boldsymbol{\Omega}$, for which each element can be viewed as an equivalent class of DNN models. Let $\nu(\boldsymbol{\gamma}, \boldsymbol{\theta}) \in \boldsymbol{\Omega}$ be an operator that maps any neural network to $\boldsymbol{\Omega}$ via appropriate transformations such as nodes permutation, sign changes, weight rescaling, etc. To serve the purpose of structure selection in the space $\boldsymbol{\Omega}$, they adopted the marginal posterior inclusion probability approach (Liang et al., 2013).

For each connection $c_i$, they defined its marginal posterior inclusion probability by

$$q_i = \int \sum_{\boldsymbol{\gamma}} e_{i|\nu(\boldsymbol{\gamma},\boldsymbol{\theta})} \pi(\boldsymbol{\gamma}|\boldsymbol{\theta}) \pi(\boldsymbol{\theta}|D_n) d\boldsymbol{\theta}, \quad i = 1, 2, \ldots, K_n, \tag{A7}$$

where $e_{i|\nu(\boldsymbol{\gamma},\boldsymbol{\theta})}$ is the indicator for the existence of connection $c_i$ in the network $\nu(\boldsymbol{\gamma}, \boldsymbol{\theta})$. Similarly, $e_{i|\nu(\boldsymbol{\gamma}^*,\boldsymbol{\theta}^*)}$ denotes the indicator for the connection $c_i$ in the true model $\nu(\boldsymbol{\gamma}^*, \boldsymbol{\theta}^*)$. Let $p_{\mu^*}$ denote the underlying true data distribution, and let $p_{\boldsymbol{\theta}}$ denote the data distribution reconstructed by the Bayesian DNN based on its posterior samples. Let $A(\epsilon_n) = \{\boldsymbol{\theta} : d(p_{\boldsymbol{\theta}}, p_{\mu^*}) \geq \epsilon_n\}$, where $d(p_1, p_2)$ denotes the Hellinger distance between two distributions. Define

$$\rho(\epsilon_n) = \max_{1 \leq i \leq K_n} \int_{A(\epsilon_n)^c} \sum_{\boldsymbol{\gamma}} |e_{i|\nu(\boldsymbol{\gamma},\boldsymbol{\theta})} - e_{i|\nu(\boldsymbol{\gamma}^*,\boldsymbol{\theta}^*)}| \pi(\boldsymbol{\gamma}|\boldsymbol{\theta}) \pi(\boldsymbol{\theta}|D_n) d\boldsymbol{\theta},$$

which measures the structure difference between the true model and the sampled models on the set $A(\epsilon_n)^c$. Further, they made the assumption:

**Assumption A6.** $\rho(\epsilon_n) \to 0$, *as $n \to \infty$ and $\epsilon_n \to 0$.*

That is, when $n$ is sufficiently large, if a DNN has approximately the same probability distribution as the true DNN, then the structure of the DNN, after mapping into the parameter space $\boldsymbol{\Omega}$, must coincide with that of the true DNN.

Let $\hat{\boldsymbol{\gamma}}_\zeta = \{i : q_i > \zeta, i = 1, 2, \ldots, K_n\}$ as an estimator of $\boldsymbol{\gamma}_* = \{i : e_{i|\nu(\boldsymbol{\gamma}^*,\boldsymbol{\theta}^*)} = 1, i = 1, \ldots, K_n\}$, where $\boldsymbol{\gamma}_*$ can be viewed as the uniquenized true model. In summary, they have the following result regarding network structure selection:

**Lemma A3.** *(Theorem 2.2; Sun et al. (2022)) Suppose Assumptions A4-A6 hold. Then, as $n \to \infty$, we have*

(a) *(sure screening)* $P(\boldsymbol{\gamma}_* \subset \hat{\boldsymbol{\gamma}}_\zeta) \xrightarrow{p} 1$ *for any pre-specified $\zeta \in (0, 1)$.*

(b) *(Consistency)* $P(\boldsymbol{\gamma}_* = \hat{\boldsymbol{\gamma}}_{0.5}) \xrightarrow{p} 1$.

As shown by Sun et al. (2022), Lemma A3 implies consistency of covariate selection.

For binary classification problems, the method leads to a predictive distribution $\hat{p}(\boldsymbol{x}) := \int p(\boldsymbol{x}; \boldsymbol{\theta}) d\pi(\boldsymbol{\theta}|D_n)$ as the Bayesian estimator of the true classification distribution $p^*(\boldsymbol{x}) := P(y = 1|\boldsymbol{x} = 1)$. Let $\hat{\mu}_A(\boldsymbol{x}) = \int A\hat{p}(\boldsymbol{x}) \nu_A(dA) = \hat{p}(\boldsymbol{x})$, and $\mu_A^*(\boldsymbol{x}) = \int A p^*(\boldsymbol{x}) \nu_A(dA) = p^*(\boldsymbol{x})$.

**Lemma A4.** *Suppose Assumptions A4-A5 hold. Then the following inequality holds with probability greater than $1 - 2\exp\{cn\epsilon_n^2\}$,*

$$\mathbb{E}_{\boldsymbol{x}}([\hat{\mu}_A(\boldsymbol{x}) - \mu_A^*(\boldsymbol{x})]^2) \leq 4\epsilon_n^2 + 16e^{-cn\epsilon_n^2/16}/\xi \asymp \epsilon_n^2,$$

*where $\xi \leq \pi(p > 0.5|D_n)$ denotes a lower bound of the selection probability of the selection rule $p > 0.5$, and $\mathbb{E}_{\boldsymbol{x}}[\cdot]$ denotes expectation with respect to $\nu_{\boldsymbol{x}}$, i.e., the probability measure of $\boldsymbol{x}$.*

*Proof.* The proof of Lemma A4 follows from the arguments of Jiang (2007) (around equations (23)-(25)), i.e., for binary classification, we have

$$\mathbb{E}_{\boldsymbol{x}}([\hat{\mu}_A(\boldsymbol{x}) - \mu_A^*(\boldsymbol{x})]^2) \leq 4d^2(\hat{p}(\boldsymbol{x}), p_{\mu^*}) \leq 4\epsilon_n^2 + 8\pi[d(p_{\boldsymbol{\theta}}, p_{\mu^*}) > \epsilon_n|D_n]/\xi.$$

The proof can then be completed by applying the first inequality of (A6) to $\pi[d(p_{\boldsymbol{\theta}}, p_{\mu^*}) > \epsilon_n|D_n]$.

Note that the overlapping condition generally assumed for the treatments ensures that $\zeta$ is bounded away from zero and thus the approximation $4\epsilon_n^2 + 16e^{-cn\epsilon_n^2/16}/\xi \asymp \epsilon_n^2$ holds. $\qquad\square$

Regarding the generalization error for regression problems, they proved the following result:

**Lemma A5.** *(Theorem 2.6; Sun et al. (2022)) Suppose Assumptions A4-A5 hold. If $\Theta$ is compact, the activation function $\psi(\cdot) \in [-1, 1]$, and $\mu^*(\boldsymbol{x})$ is bounded, then the following inequality holds with probability greater than $1 - 2\exp\{cn\epsilon_n^2\}$,*

$$\mathbb{E}_{\boldsymbol{x}}(\int \mu(\boldsymbol{\theta}, \boldsymbol{x})\pi(\boldsymbol{\theta}|D_n)d\boldsymbol{\theta} - \mu^*(\boldsymbol{x}))^2 \asymp (\epsilon_n^2 + e^{-cn\epsilon_n^2})\overline{L}_n^2,$$

*where $\epsilon_n$ is as defined in Lemma A2, and $\mathbb{E}_{\boldsymbol{x}}(\cdot)$ denotes an expectation with respect to $\nu(\boldsymbol{x})$, the probability measure of $\boldsymbol{x}$.*

To accelerate computation, Sun et al. (2022) suggested to replace the Bayesian estimators involved in Lemmas A3-A5 by their Laplace approximators. That is, instead of sampling from the posterior distribution, they conducted optimization to maximize the objective function

$$L_n(\boldsymbol{\theta}) = \frac{1}{n}\sum_{i=1}^n \log(p(y_i, \boldsymbol{x}_i; \boldsymbol{\theta})) + \frac{1}{n}\log(\pi(\boldsymbol{\theta})), \tag{A8}$$

where $\pi(\boldsymbol{\theta})$ denotes the mixture Gaussian prior as specified in (8). Denote the resulting maximum *a posteriori* (MAP) estimator by

$$\hat{\boldsymbol{\theta}}_n = \arg\max_{\boldsymbol{\theta} \in \Theta} L_n(\boldsymbol{\theta}). \tag{A9}$$

**Assumption A7.** *Assume $r_n = |\boldsymbol{\gamma}^*|$ grows with the sample size $n$ at a rate of $o(n^{1/4})$, i.e. $r_n \prec n^{1/4}$.*

By invoking the Laplace approximation theorem, they show that the consistency results established in Lemmas A3-A5 still hold for $\hat{\boldsymbol{\theta}}_n$ with the approximation error decaying at a rate of $O(\frac{r_n^4}{n})$. This leads to the following corollary:

**Corollary 1.** *For sparse DNNs, the consistency results established in Lemmas A3-A5 also hold for the maximum a posteriori estimator (A9), provided that Assumption A7 also holds.*

Moreover, to address the local trap issue possibly encountered in maximizing (A8), Sun et al. (2022) suggested to run an optimization procedure, such as Adam or SGD, multiple times and select the solution according to the BIC criterion.

**Proof of Theorem 1**

*Proof.* First, we note that these results hold for the sparse DNN following from Lemmas A3-A5 and the property of Laplace approximation. Specifically, the term $o(n^{-1/2})$ in parts (a) and (b) follows from Theorem 2.3 of Sun et al. (2022), where accuracy of the Laplace approximation for Bayesian sparse DNNs is given as $O(r_n^4/n)$. Therefore, for the mean squared errors in part (a), we have

$$\mathbb{E}_{\boldsymbol{x}}[(\hat{p}(\boldsymbol{x}; \hat{\boldsymbol{\theta}}_n) - p^*(\boldsymbol{x}))^2] \leq 2\mathbb{E}_{\boldsymbol{x}}[(\hat{p}(\boldsymbol{x}) - p^*(\boldsymbol{x}))^2] + 2\mathbb{E}_{\boldsymbol{x}}[(\hat{p}(\boldsymbol{x}; \hat{\boldsymbol{\theta}}_n) - \hat{p}(\boldsymbol{x}))^2]$$
$$= O\left(\epsilon_n^2 + e^{-cn\epsilon_n^2/16}\right) + O(r_n^8/n^2)$$

where $\hat{p}(\boldsymbol{x}) = \int p(\boldsymbol{x}; \boldsymbol{\theta}) d\pi(\boldsymbol{\theta}|D_n)$, as defined in Lemma A4, denotes the Bayesian estimator of $p^*(\boldsymbol{x})$. Therefore, the result follows under the assumption $r_n \prec n^{3/16}$. For part (b), the result can be justified in a similar way.

Then, by equation (5), these results can also be achieved by the Causal-StoNet which is trained by maximizing the penalized log-likelihood function (9). This completes the proof of the theorem. $\qquad\square$

## A.5 Proof of Theorem 2

For convenience, for any $a, a' \in \{0, 1\}$, we define the following notations:

$$
\begin{aligned}
p_a(\boldsymbol{x}) &= \mathbb{E}(P(A = a|\boldsymbol{X} = \boldsymbol{x})), \quad p_a = \mathbb{E}_{\boldsymbol{x}}(p_a(\boldsymbol{x})), \\
\mu_a(\boldsymbol{x}) &= \mathbb{E}(Y|A = a, \boldsymbol{X} = \boldsymbol{x}), \quad \mu_a = \mathbb{E}(Y(a)), \\
\sigma_a^2(\boldsymbol{x}) &= \mathbb{E}(U^2|A = a, \boldsymbol{X} = \boldsymbol{x}).
\end{aligned}
$$

With the above notation, we have

$$
V_\tau = \mathbb{E}\left[\frac{\sigma_1^2(\boldsymbol{X})}{p_1(\boldsymbol{X})} + \frac{\sigma_0^2(\boldsymbol{X})}{p_0(\boldsymbol{X})}\right] + \mathbb{E}\left[((\mu_1(\boldsymbol{X}) - \mu_1) - (\mu_0(\boldsymbol{X}) - \mu_0))^2\right].
$$

For estimation, we define

$$
\hat{\mu}_a(\boldsymbol{x}) = \hat{\mu}(\boldsymbol{X} = \boldsymbol{x}, A = a, \hat{\boldsymbol{\theta}}_n), \quad \hat{p}_a(\boldsymbol{x}) = \hat{p}(\boldsymbol{X} = \boldsymbol{x}, A = a, \hat{\boldsymbol{\theta}}_n), \quad \hat{p}_a = \frac{1}{n}\sum_{i=1}^n I(A_i = a),
$$

$$
\hat{\mu}_a = \frac{1}{n}\sum_{i=1}^n \left\{\frac{I(A_i = a)(y_i - \hat{\mu}_a(\boldsymbol{x}_i))}{\hat{p}_a(\boldsymbol{x}_i)} + \hat{\mu}_a(\boldsymbol{x}_i)\right\},
$$

With the notation, we have

$$
\hat{V}_\tau = \mathbb{E}_n\left[\frac{I(A_i = 1)(y_i - \hat{\mu}_1(\boldsymbol{x}_i))^2}{\hat{p}_1(\boldsymbol{x}_i)^2} + \frac{I(A_i = 0)(y_i - \hat{\mu}_0(\boldsymbol{x}_i))^2}{\hat{p}_0(\boldsymbol{x}_i)^2}\right] + \mathbb{E}_n\left[((\hat{\mu}_1(\boldsymbol{x}_i) - \hat{\mu}_1)) - (\hat{\mu}_0(\boldsymbol{x}_i) - \hat{\mu}_0))^2\right],
$$

where $\mathbb{E}[\cdot]$ denotes the empirical mean over $n$ samples.

**Proof of Theorem 2**

*Proof.* The results directly follow from Theorem 3 and Corollary 2 of Farrell (2015). It is easy to verify that the conditions $n^{-1+\xi} \prec \varpi_n^2 \prec n^{-\frac{1}{2}-\xi}$, $0.5 + \xi < \varepsilon < 1 - \xi$, and $\overline{L}_n = O(n^\xi)$ ensure that

$$
\epsilon_n^2 + e^{-cn\epsilon_n^2/16} \prec n^{-1/2}, \quad \text{and} \quad (\epsilon_n^2 + e^{-cn\epsilon_n^2})\overline{L}_n^2 \prec n^{-1/2},
$$

and thus Assumption 3 of Farrell (2015) holds. Assumptions 1 and 2 of Farrell (2015) are given in Assumption A1 of this paper. Therefore, Theorem 3 and Corollary 2 of Farrell (2015) hold, and the statement of Theorem 2 is implied. $\qquad\square$

## A.6 Convergence of the SGHMC Algorithm

*Notations:* We let $\boldsymbol{D}$ denote a dataset of $n$ observations, and let $D_i$ denote the $i$-th observation of $\boldsymbol{D}$. For StoNet, $D_i$ has included both the input and output variables of the observation. For simplicity of notation, we re-denote the latent variable corresponding to $D_i$ by $Z_i$, and denote by $f_{D_i}(z_i, \boldsymbol{\theta}) = -\log \pi(z_i|D_i, \boldsymbol{\theta})$ the negative log-density function of $Z_i$. Let $\boldsymbol{z} = (z_1, z_2, \ldots, z_n)$ be a realization of $\boldsymbol{Z} = (Z_1, Z_2, \ldots, Z_n)$, and let $F_{\boldsymbol{D}}(\boldsymbol{z}, \boldsymbol{\theta}) = \sum_{i=1}^n f_{D_i}(z_i, \boldsymbol{\theta})$.

To study the convergence of Algorithm 1, we need the following assumptions:

**Assumption A8.** *The function $F_{\boldsymbol{D}}(\cdot, \cdot)$ takes nonnegative real values, and there exist constants $A, B \geq 0$, such that $|F_{\boldsymbol{D}}(\boldsymbol{0}, \boldsymbol{\theta}^*)| \leq A$, $\|\nabla_{\boldsymbol{Z}} F_{\boldsymbol{D}}(\boldsymbol{0}, \boldsymbol{\theta}^*)\| \leq B$, $\|\nabla_{\boldsymbol{\theta}} F_{\boldsymbol{D}}(\boldsymbol{0}, \boldsymbol{\theta}^*)\| \leq B$, and $\|H(\boldsymbol{0}, \boldsymbol{\theta}^*)\| \leq B$.*

**Assumption A9.** *(Smoothness) $F_{\boldsymbol{D}}(\cdot, \cdot)$ is $M$-smooth and $H(\cdot, \cdot)$ is $M$-Lipschitz: there exists some constant $M > 0$ such that for any $\boldsymbol{Z}, \boldsymbol{Z}' \in \mathbb{R}^{d_z}$ and any $\boldsymbol{\theta}, \boldsymbol{\theta}' \in \Theta$,*

$$\|\nabla_{\boldsymbol{Z}} F_{\boldsymbol{D}}(\boldsymbol{Z}, \boldsymbol{\theta}) - \nabla_{\boldsymbol{Z}} F_{\boldsymbol{D}}(\boldsymbol{Z}', \boldsymbol{\theta}')\| \le M\|\boldsymbol{Z} - \boldsymbol{Z}'\| + M\|\boldsymbol{\theta} - \boldsymbol{\theta}'\|,$$
$$\|\nabla_{\boldsymbol{\theta}} F_{\boldsymbol{D}}(\boldsymbol{Z}, \boldsymbol{\theta}) - \nabla_{\boldsymbol{\theta}} F_{\boldsymbol{D}}(\boldsymbol{Z}', \boldsymbol{\theta}')\| \le M\|\boldsymbol{Z} - \boldsymbol{Z}'\| + M\|\boldsymbol{\theta} - \boldsymbol{\theta}'\|,$$
$$\|H(\boldsymbol{Z}, \boldsymbol{\theta}) - H(\boldsymbol{Z}', \boldsymbol{\theta}')\| \le M\|\boldsymbol{Z} - \boldsymbol{Z}'\| + M\|\boldsymbol{\theta} - \boldsymbol{\theta}'\|.$$

**Assumption A10.** *(Dissipativity) For any $\boldsymbol{\theta} \in \Theta$, the function $F_{\boldsymbol{D}}(\cdot, \boldsymbol{\theta}^*)$ is $(m, b)$-dissipative: there exist some constants $m > \frac{1}{2}$ and $b \ge 0$ such that $\langle \boldsymbol{Z}, \nabla_{\boldsymbol{Z}} F_{\boldsymbol{D}}(\boldsymbol{Z}, \boldsymbol{\theta}^*) \rangle \ge m\|\boldsymbol{Z}\|^2 - b$.*

The smoothness and dissipativity conditions are regular for studying the convergence of stochastic gradient MCMC algorithms, and they have been used in many papers such as Raginsky et al. (2017) and Gao et al. (2021). As implied by the definition of $F_{\boldsymbol{D}}(\boldsymbol{z}, \boldsymbol{\theta})$, the values of $M$, $m$ and $b$ increase linearly with the sample size $n$. Therefore, we can impose a nonzero lower bound on $m$ to facilitate related proofs.

**Assumption A11.** *(Gradient noise) There exists a constant $\varsigma \in [0, 1)$ such that for any $\boldsymbol{Z}$ and $\boldsymbol{\theta}$, $\mathbb{E}\|\nabla_{\boldsymbol{Z}} \hat{F}_{\boldsymbol{D}}(\boldsymbol{Z}, \boldsymbol{\theta}) - \nabla_{\boldsymbol{Z}} F_{\boldsymbol{D}}(\boldsymbol{Z}, \boldsymbol{\theta})\|^2 \le 2\varsigma(M^2\|\boldsymbol{Z}\|^2 + M^2\|\boldsymbol{\theta} - \boldsymbol{\theta}^*\|^2 + B^2)$.*

Introduction of the extra constant $\varsigma$ facilitates our study. For the full data case, we have $\varsigma = 0$, i.e., the gradient $\nabla_{\boldsymbol{Z}} F_{\boldsymbol{D}}(\boldsymbol{Z}, \boldsymbol{\theta})$ can be evaluated accurately.

**Assumption A12.** *The step size $\{\gamma_k\}_{k \in \mathbb{N}}$ is a positive decreasing sequence such that $\gamma_k \to 0$ and $\sum_{k=1}^{\infty} \gamma_k = \infty$. In addition, let $h(\boldsymbol{\theta}) = \mathbb{E}(H(\boldsymbol{Z}, \boldsymbol{\theta}))$, then there exists $\delta > 0$ such that for any $\boldsymbol{\theta} \in \Theta$, $\langle \boldsymbol{\theta} - \boldsymbol{\theta}^*, h(\boldsymbol{\theta})) \rangle \ge \delta\|\boldsymbol{\theta} - \boldsymbol{\theta}^*\|^2$, and $\liminf_{k \to \infty} 2\delta \frac{\gamma_k}{\gamma_{k+1}} + \frac{\gamma_{k+1} - \gamma_k}{\gamma_{k+1}^2} > 0$.*

As shown by Benveniste et al. (1990) (p.244), Assumption A12 can be satisfied by setting $\gamma_k = \tilde{a}/(\tilde{b} + k^\alpha)$ for some constants $\tilde{a} > 0$, $\tilde{b} \ge 0$, and $\alpha \in (0, 1 \wedge 2\delta\tilde{a})$. By (A2), $\delta$ increases linearly with the sample size $n$. Therefore, if we set $\tilde{a} = \Omega(1/n)$ then $2\delta\tilde{a} > 1$ can be satisfied, where $\Omega(\cdot)$ denotes the order of the lower bound of a function. In this paper, we simply choose $\alpha \in (0, 1)$ by assuming that $\tilde{a}$ has been set appropriately with $2\delta\tilde{a} \ge 1$ held.

**Assumption A13.** *(Solution of Poisson equation) For any $\boldsymbol{\theta} \in \Theta$, $\boldsymbol{z} \in \mathfrak{Z}$, and a function $V(\boldsymbol{z}) = 1 + \|\boldsymbol{z}\|$, there exists a function $\mu_{\boldsymbol{\theta}}$ on $\mathfrak{Z}$ that solves the Poisson equation $\mu_{\boldsymbol{\theta}}(\boldsymbol{z}) - \mathcal{T}_{\boldsymbol{\theta}}\mu_{\boldsymbol{\theta}}(\boldsymbol{z}) = H(\boldsymbol{\theta}, \boldsymbol{z}) - h(\boldsymbol{\theta})$, where $\mathcal{T}_{\boldsymbol{\theta}}$ denotes a probability transition kernel with $\mathcal{T}_{\boldsymbol{\theta}}\mu_{\boldsymbol{\theta}}(\boldsymbol{z}) = \int_{\mathfrak{Z}} \mu_{\boldsymbol{\theta}}(\boldsymbol{z}')\mathcal{T}_{\boldsymbol{\theta}}(\boldsymbol{z}, \boldsymbol{z}')d\boldsymbol{z}'$, such that*

$$H(\boldsymbol{\theta}_k, \boldsymbol{z}_{k+1}) = h(\boldsymbol{\theta}_k) + \mu_{\boldsymbol{\theta}_k}(\boldsymbol{z}_{k+1}) - \mathcal{T}_{\boldsymbol{\theta}_k}\mu_{\boldsymbol{\theta}_k}(\boldsymbol{z}_{k+1}), \quad k = 1, 2, \dots. \tag{A10}$$

*Moreover, for all $\boldsymbol{\theta}, \boldsymbol{\theta}' \in \Theta$ and $\boldsymbol{z} \in \mathfrak{Z}$, we have $\|\mu_{\boldsymbol{\theta}}(\boldsymbol{z}) - \mu_{\boldsymbol{\theta}'}(\boldsymbol{z})\| \le \varsigma_1\|\boldsymbol{\theta} - \boldsymbol{\theta}'\|V(\boldsymbol{z})$ and $\|\mu_{\boldsymbol{\theta}}(\boldsymbol{z})\| \le \varsigma_2 V(\boldsymbol{z})$ for some constants $\varsigma_1 > 0$ and $\varsigma_2 > 0$.*

This assumption is also regular for studying the convergence of stochastic gradient MCMC algorithms, see e.g., Whye et al. (2016) and Deng et al. (2019). Alternatively, one can assume that the MCMC algorithms satisfy the drift condition, and then Assumption A13 can be verified, see e.g., Andrieu et al. (2005).

**Outline of the Proof of Lemma 1** Lemma 1 can be proved in a similar way to Theorem 3.1 of Liang et al. (2022). Note that in the proof of Lemma 1, the boundedness of $\Theta$ is not assumed.

**Outline of the Proof of Lemma 2** Lemma 2 can be proved in a similar way to Theorem 3.3 of Liang et al. (2014) by ignoring the parameter $\tau$ used in the proof there and modifying some notations appropriately.

## A.7 More Numerical Results

### A.7.1 An Illustrative Example with Varying Sample Size

**Data Generation Procedure** 10 simulation datasets are generated in the following procedure, which is inspired by Lei & Candès (2021).

- Generate $e, z_1, \cdots, z_{1000}$ independently from a truncated standard normal distribution on the interval $[-10, 10]$. Set $x_i = \frac{e + z_i}{\sqrt{2}}$ for $i = 1, \cdots, 1000$, making the covariates highly correlated.
- The propensity score $e(\boldsymbol{x}) = \frac{1}{4}(1 + \beta_{2,4}(\frac{1}{3}(\Phi(x_1) + \Phi(x_3) + \Phi(x_5))))$, where $\beta_{2,4}$ is the CDF of the beta distribution with shape parameters $(2, 4)$, and $\Phi$ denotes the CDF of the standard normal distribution. This ensures that $e(x) \in [0.25, 0.5]$, thereby providing sufficient overlap. Treatment $A_i$ is hence generated by a Bernoulli distibution with the probablity of success being $e(x_i)$, and resampling from the treatment and control groups has been performed for ensuring that the dataset contains balanced samples for treatment group and control group.
- For simulation of observed outcome, we consider

$$y_i = c(\boldsymbol{x}_i) + \tau A_i + \eta_i * A_i + \sigma z_i, \quad i = 1, 2, \ldots, n,$$
$$c(\boldsymbol{x}_i) = \frac{5x_3}{1 + x_4^2} + 2x_5$$

where $\eta(x_i) = f(x_1)f(x_2) - E(f(x_1)f(x_2))$ and $f(x) = \frac{2}{1 + \exp(-x + 0.5))}$. In other words, we set treatment effect $\tau(x_i) = \tau + \eta_i$. We generated the data under the setting $\tau = 3$ and $\sigma = 0.25$ with $n_{train} \in \{800, 1600, 2400, 3200, 4000\}$, $n_{val} \in \{200, 400, 600, 800, 1000\}$, $n_{test} \in \{200, 400, 600, 800, 1000\}$.

**Results**  For TMLE, we use the ensemble of lasso and XGBoost to estimate the nuisance functions. For X-Learner, we consider the model with Random Forest (X-Learner-RF) and the model with Bayesian Additive Regression Trees (X-Learner-BART), but only presents the result of X-Learner-RF, since the performance of X-Learner-RF is consistently better than that of X-Learner-BART on our simulation dataset. For CFRNet, we consider the model with Wasserstein distance (CFRNet-Wass) and the model with Maximum Mean Discrepancy (CFRNet-MMD), and only presents the result of CFRNet-MMD, following a similiar logic. For most of the benchmark models, we refer to the code implementation of (Jiang et al., 2023). The results are summarzed in 2.

### A.7.2 An Illustrative Example for Missing Data Problems

**Data Generation Procedure**  With the following procedure, we simulated 10 datasets for three scenarios: a) complete data, b) missing at random (MAR), and c) missing not at random (MNAR). Each dataset consists of 10,000 training samples, 1000 validation samples, and 1000 test samples. In our setting, only training set contains missing values.

1. Generate $x_1, \cdots, x_{100}$ from an auto-regressive process of order 2 with the concentration matrix given by

$$C_{i,j} = \begin{cases} 0.5, & \text{if } |j - i| = 1, i = 2, \cdots, 99. \\ 0.25, & \text{if } |j - i| = 2, i = 3, \cdots, 98. \\ 1, & \text{if } j = i, i = 1, \cdots, 100. \\ 0, & \text{otherwise} \end{cases}$$

2. Generate the binary treatment variable $A \in \{0, 1\}$ from a Bernoulli distribution with the success probability given by $p(x_1, x_2, x_3, x_5) = 1/(1 + e^{-s(x_1, x_2, x_3, x_5)})$, where

$$s(x_1, x_2, x_3, x_5) = \tanh(-x_1 - 2x_5) - \tanh(2x_2 - 2x_3),$$

and resampling from the treatment and control groups has been performed for ensuring that the dataset contains equal numbers of treatment and control samples.

3. Generate outcome variable $Y$ by

$$\begin{aligned} Y = &- 4\tanh(\tanh(-2\tanh(2x_1 + x_4) + \tanh(2x_2 - 2x_3) - 2A) \\ &+ 2\tanh(-A + 2\tanh(\tanh(2x_2 - 2x_3) - 2\tanh(-2x_4 + x_5))) \\ &+ 0x_6 + \cdots + 0x_{100} + \epsilon \end{aligned}$$

where $\epsilon \sim N(0, 1)$ and is independent of $x_i$'s.

4. For MAR, we randomly deleted 10% of the observations in $x_1$ and $x_4$ as missing values. For MNAR, we first generate missing pattern $R_1$ and $R_4$, which are binary vectors with 1 representing observed and 0 representing missing, from Bernoulli distribution with the success probability given by $p_1 = 1/(1 + e^{-s_1(x_1,\cdots,x_{100})})$ and $p_4 = 1/(1 + e^{-s_4(x_1,\cdots,x_{100})})$, where

$$s_1(x_1,\cdots,x_{100}) = 4 - 2A + (-0.1)^{j-1}x_{2j-1}, \text{ where } j = 1,\cdots,50$$

and

$$s_4(x_1,\cdots,x_{100}) = 4 - 2A + (-0.1)^{j}x_{2j}, \text{ where } j = 1,\cdots,50$$

then delete the observations based on $R_1$ and $R_4$. The missing rate for MNAR scenario is roughly 10%.

**Definition of the False Selection Rate (FSR) and Negative Selection Rate (NSR)**
In this paper, the FSR and NSR are defined based on 10 datasets:

$$\text{FSR} = \frac{\sum_{i=1}^{10}|\hat{S}_i \setminus S|}{\sum_{i=1}^{10}\hat{S}_i}, \quad \text{NSR} = \frac{\sum_{i=1}^{10}|S \setminus \hat{S}_i|}{\sum_{i=1}^{10}S},$$

where $S$ is the set of true variables, $S_i$ is the set of selected variables for dataset $i$, and $|S_i|$ is the size of $S_i$.

**Results**   For this experiment, we assume that we already know the neighborhood structure of the Gaussian graphical model that represents the correlations between covariates, rendering the distribution $\pi(\boldsymbol{X}_{mis}|\boldsymbol{X}_{obs})$ be correctly modeled. For case where the structure is unknown, estimating the structure is necessary before running the Causal-StoNet. The covariate selection accuracy and Out-of-Sample MAE of the estimated ATE are summarized in Table A1, as compared with the result for complete dataset. The result shows that even with missing values present in the dataset, Causal-StoNet can still correctly identify the true covariates for the outcome and propensity score models and achieve reasonably accurate estimates for the ATE.

Table A1:   Covariate selection accuracy and MAE of the ATE estimates for the illustrative example of missing data.

|  | Out-of Sample MAE | $FSR_Y$ | $NSR_Y$ | $FSR_A$ | $NSR_A$ |
|---|---|---|---|---|---|
| Complete | 0.0103(0.0024) | 0 | 0 | 0 | 0 |
| MAR | 0.0797(0.0134) | 0 | 0 | 0 | 0 |
| MNAR | 0.1687(0.0284) | 0 | 0 | 0 | 0 |

### A.7.3   TWINS DATA

**Data Preprocessing**   In data preprocessing, we focused on the same-sex twin pairs with born-weights less than 2 kg and assigned treatments according to $t_i|x_i, z_i \sim \text{Bernoulli}(\sigma(w_o^T \boldsymbol{x} + w_h(z/10 - 0.1)))$. Here, $\sigma(\cdot)$ is sigmoid, $w_o \sim N(0, 0.1 \cdot I)$, $w_h \sim N(5, 0.1)$, $z$ represents the covariate **gestat10** (gestation weeks before birth), and $\boldsymbol{x}$ encompasses the other 45 covariates.

### A.7.4   TCGA-BRCA DATA

The Breast Cancer dataset from the TCGA database collects clinical data and gene expression data for breast cancer patients. The gene expression data contains the expression measurements of 20531 genes. Our goal is to investigate the causal effect of radiation therapy on patients' vital status, while accounting for relevant clinical features and genetic covariates.

The treatment variable $A$ is binary (1 for radiation therapy, 0 otherwise), and the outcome variable $Y$ is also binary (1 for death, 0 otherwise). For genetic covariates, we applied the sure independence variable screening method (Cui et al., 2015) to reduce the number of genes from 20531 to 100. After preprocessing, the dataset contains 110 covariates with a

Table A2: Covariates selected for the treatment and outcome models by the Causal-StoNet in a three-fold cross-validation experiment for the twins data, where the number in the parentheses indicates the times that the covariate was elected.

| | |
|---|---|
| Treatment Model | pldel(1), birattnd(1), mager8(3), ormoth(3), mrace(3) meduc6(3), dmar(1), mpre5(3), adequacy(3), frace(3), birmon(3) gestat10(3), csex(1), incervix(2), cigar6(3), crace(3), data__year(3) nprevistq(3), dfageq(3), feduc6(3), dlivord_min(3), dtotord_min(3), brstate_reg(3), stoccfipb_reg(3), mplbir_reg(3), bord(3) |
| Outcome Model | pldel(3), birattnd(3), mager8(3), ormoth(3), mrace(3) meduc6(3), dmar(2), mpre5(3), adequacy(3), frace(3), birmon(3), gestat10(3), csex(3), hydra(2), incervix(3), cigar6(3), crace(3), data__year(3) nprevistq(3), dfageq(3), feduc6(3), dlivord_min(3), dtotord_min(3) brstate_reg(3), stoccfipb_reg(3), mplbir_reg(3), bord(3) |

Table A3: ATE estimates by different methods for BRCA-TCGA data

| Causal-StoNet | DSE | ARBE | TMLE(Lasso) | TMLE(ensemble) | DONUT |
|---|---|---|---|---|---|
| -0.1243 (0.0058) | -0.0378(0.0243) | -0.1497 (0.1021) | -0.0509 (0.0172) | -0.0468 (0.0153) | -0.0129 ( 0.0088) |

sample size of 845. Table A3 reports the ATE estimates by different methods, with the Causal-StoNet estimate having a smaller standard error. Table A4 summarizes the most frequently selected covariates for both the treatment and outcome models across the three cross-validation folds. Our method identified **pathology N stage** and gene **ALDH3A2** as significant variables. It is exciting to note that **clinical N stage** and the tumor expression of **ALDH3A2** have been reported by Xie et al. (2022) as potential markers for predicting tumor recurrence in breast cancer patients who achieve a pathologic complete response after neoadjuvant chemotherapy.

Table A4: Most frequently selected covariates for both the treatment and outcome models by the Causal-StoNet in a three-fold cross-validation experiment for the TCGA-BRCA data

| | |
|---|---|
| Clinical Features | years to birth, date of initial pathologic diagnosis, number of lymph nodes, pathology N stage[†] |
| Genetic Features | ACMSD, AKAP2, ALDH3A2[†], ALG1, ALMS1P, ANKLE1, ANKRD54, AQP7P1, AZI1, B4GALT1, B4GALT2, BCL6B, C10orf47, C10orf72 |

## A.8 Parameters Setting Used in the Experiments

### A.8.1 Causal-StoNet

**Illustrated Example with Varying Sample Size** The network has 3 hidden layers with structure 32-16-8-1, where Tanh is used as the activation function. The treatment variable is set at the second hidden node of the second hidden layer. Variances of latent variables are $\sigma_{n,1}^2 = 10^{-3}$, $\sigma_{n,2}^2 = 10^{-5}$, $\sigma_{n,3}^2 = 10^{-7}$, $\sigma_{n,4}^2 = 10^{-9}$. For the mixture Gaussian prior, $\sigma_0^2 = 10^{-5}$, $\sigma_1^2 = 0.01$, and $\lambda_n = 10^{-6}$. The epochs for pre-training, training, and refining after sparsification are 50, 200, and 200, respectively.

For SGHMC imputation, $t_{HMC} = 1$, $\alpha = 0.1$. Initial imputation learning rate is set as $\epsilon_1 = 3 \times 10^{-3}$, $\epsilon_2 = 3 \times 10^{-4}$, $\epsilon_3 = 5 \times 10^{-7}$, and decays at training stage and refining stage. For epoch $k$ in these stages, the imputation learning rate is $\epsilon_{k,i} = \frac{\epsilon_i}{1+\epsilon_i \times k^{1.2}}$.

For parameter update, Initial learning rate for training stage are $\gamma_1 = 10^{-3}$, $\gamma_2 = 10^{-6}$, $\gamma_3 = 10^{-8}$, $\gamma_4 = 5 \times 10^{-13}$, and the initial learning rate for refining stage after the sparsification are $\gamma_1 = 10^{-4}$, $\gamma_2 = 10^{-7}$, $\gamma_3 = 10^{-9}$, $\gamma_4 = 5 \times 10^{-14}$. Learning rate decays for training and refining stage, and for epoch $k$ in these stages, the learning rate is $\gamma_{k,i} = \frac{\gamma_i}{1+\gamma_i \times k^{1.4}}$.

**Illustrated Example with Missing Value** The network has 3 hidden layers with structure 8-6-5-1, where Tanh is used as the activation function. The treatment variable is set at the second hidden node of the second hidden layer. Variances of latent variables are $\sigma_{n,1}^2 = 10^{-3}$, $\sigma_{n,2}^2 = 10^{-5}$, $\sigma_{n,3}^2 = 10^{-7}$, $\sigma_{n,4}^2 = 10^{-9}$. For the mixture Gaussian prior, $\sigma_0^2 = 3 \times 10^{-3}$, $\sigma_1^2 = 0.3$, and $\lambda_n = 10^{-6}$. The epochs for pre-training, training, and refining after sparsification are 100, 1500, and 200, respectively.

For SGHMC imputation, $t_{HMC} = 1$, $\alpha = 0.1$. Initial imputation learning rate for the network is set as $\epsilon_1 = 3 \times 10^{-3}$, $\epsilon_2 = 3 \times 10^{-4}$, $\epsilon_3 = 10^{-6}$, and imputation learning rate for missing covariates imputation $\epsilon_{\text{miss}} = 3 \times 10^{-4}$. Both imputation learning rates decay at training stage and refining stage. For epoch $k$ in these stages, the imputation learning rate is $\epsilon_{k,i} = \frac{\epsilon_i}{1 + \epsilon_i \times k^{1.2}}$.

For parameter update, Initial learning rate for training stage are $\gamma_1 = 10^{-3}$, $\gamma_2 = 3 \times 10^{-6}$, $\gamma_3 = 10^{-7}$, $\gamma_4 = 10^{-12}$, and the initial learning rate for refining stage after the sparsification are $\gamma_1 = 10^{-4}$, $\gamma_2 = 3 \times 10^{-7}$, $\gamma_3 = 10^{-8}$, $\gamma_4 = 10^{-13}$. Learning rate decays for training and refining stage, and for epoch $k$ in these stages, the learning rate is $\gamma_{k,i} = \frac{\gamma_i}{1 + \gamma_i \times k^{1.2}}$.

**ACIC** The network has 2 hidden layers with structure 200-64-32, where Tanh is used as the activation function. The treatment variable is set at the second hidden node of the first hidden layer. Variances of latent variables are $\sigma_{n,1}^2 = 10^{-5}$, $\sigma_{n,2}^2 = 10^{-7}$, $\sigma_{n,2}^2 = 10^{-9}$. For the mixture Gaussian prior, $\sigma_0^2 = 2 \times 10^{-4}$, $\sigma_1^2 = 0.1$, and $\lambda_n = 10^{-6}$. The number of epochs for pre-training is 100. The parameters were pruned twice, first after 500 training epochs, and second after 1000 training epochs. The first pruning can be deemed as another initialization for the model. After the second pruning, the model was trained for another 200 epochs for refining the network parameters.

For SGHMC imputation, $t_{HMC} = 1$, $\alpha = 1$. Initial imputation learning rate is set as $\epsilon_1 = 3 \times 10^{-4}$, $\epsilon_2 = 10^{-6}$, and decays at training stage and refining stage. For epoch $k$ in these stages, the imputation learning rate is $\epsilon_{k,i} = \frac{\epsilon_i}{1 + \epsilon_i \times k}$.

For parameter update, Initial learning rate for training stage are $\gamma_1 = 5 \times 10^{-6}$, $\gamma_2 = 5 \times 10^{-8}$, $\gamma_3 = 5 \times 10^{-13}$, and the initial learning rate for refining stage after the sparsification are $\gamma_1 = 5 \times 10^{-7}$, $\gamma_2 = 5 \times 10^{-9}$, $\gamma_3 = 10^{-13}$. Learning rate decays for training and refining stage, and for epoch $k$ in these stages, the learning rate is $\gamma_{k,i} = \frac{\gamma_i}{1 + \gamma_i \times k^2}$.

**Twins** The network has 3 hidden layers with structure 46-16-8-4, where Tanh is used as the activation function. The treatment variable is set at the second hidden node of the second hidden layer. Variances of latent variables are $\sigma_{n,1}^2 = 10^{-3}$, $\sigma_{n,2}^2 = 10^{-4}$, $\sigma_{n,3}^2 = 10^{-5}$, $\sigma_{n,4}^2 = 10^{-6}$. For the mixture Gaussian prior, $\sigma_0^2 = 5 \times 10^{-4}$, $\sigma_1^2 = 0.1$, and $\lambda_n = 10^{-6}$. The epochs for pre-training, training, and refining after sparsification are 100, 1500, and 200, respectively.

For SGHMC imputation, $t_{HMC} = 1$, $\alpha = 1$. Initial imputation learning rate is set as $\epsilon_1 = \times 10^{-2}$, $\epsilon_2 = 9 \times 10^{-3}$, $\epsilon_3 = 9 \times 10^{-5}$, and decays at training stage and refining stage. For epoch $k$ in these stages, the imputation learning rate is $\epsilon_{k,i} = \frac{\epsilon_i}{1 + \epsilon_i \times k}$.

For parameter update, Initial learning rate for training stage are $\gamma_1 = 10^{-3}$, $\gamma_2 = 10^{-4}$, $\gamma_3 = 10^{-5}$, $\gamma_4 = 10^{-9}$, and the initial learning rate for refining stage after the sparsification are $\gamma_1 = 10^{-4}$, $\gamma_2 = 10^{-5}$, $\gamma_3 = 10^{-6}$, $\gamma_4 = 10^{-10}$. Learning rate decays for training and refining stage, and for epoch $k$ in these stages, the learning rate is $\gamma_{k,i} = \frac{\gamma_i}{1 + \gamma_i \times k^{1.2}}$.

**TCGA-BRCA** The network has 3 hidden layers with structure 110-64-16-4, where Tanh is used as the activation function. The treatment variable is set at the second hidden node of the second hidden layer. Variances of latent variables are $\sigma_{n,1}^2 = 10^{-3}$, $\sigma_{n,2}^2 = 10^{-5}$, $\sigma_{n,3}^2 = 10^{-7}$, $\sigma_{n,4}^2 = 10^{-9}$. For the mixture Gaussian prior, $\sigma_0^2 = 10^{-5}$, $\sigma_1^2 = 0.01$, and $\lambda_n = 10^{-6}$. The epochs for pre-training, training, and refining after sparsification are 100, 1500, and 200, respectively.

For SGHMC imputation, $t_{HMC} = 1$, $\alpha = 1$. Initial imputation learning rate is set as $\epsilon_1 = 3 \times 10^{-3}$, $\epsilon_2 = 3 \times 10^{-4}$, $\epsilon_3 = 10^{-6}$, and decays at training stage and refining stage. For epoch $k$ in these stages, the imputation learning rate is $\epsilon_{k,i} = \frac{\epsilon_i}{1+\epsilon_i \times k^1}$.

For parameter update, Initial learning rate for training stage are $\gamma_1 = 10^{-3}$, $\gamma_2 = 5 \times 10^{-5}$, $\gamma_3 = 5 \times 10^{-7}$, $\gamma_4 = 10^{-12}$, and the initial learning rate for refining stage after the sparsification are $\gamma_1 = 10^{-4}$, $\gamma_2 = 10^{-6}$, $\gamma_3 = 10^{-8}$, $\gamma_4 = 10^{-13}$. Learning rate decays for training and refining stage, and for epoch $k$ in these stages, the learning rate is $\gamma_{k,i} = \frac{\gamma_i}{1+\gamma_i \times k^{1.2}}$.

