# OpenReview forum: "Causal-StoNet: Causal Inference for High-Dimensional Complex Data"
_ICLR.cc/2024/Conference — ICLR 2024 poster_

### Official Review · Reviewer_vAHH · 2023-10-30

**Soundness:** 4 excellent
**Presentation:** 3 good
**Contribution:** 4 excellent
**Rating:** 8
**Confidence:** 4

**Summary:**

The authors propose an algorithm, CausalStoNet, which aims to give accurate causal inferences for systems of up to 100 variables (after variable reduction) under fairly relaxed nonlinearity conditions. The method is based on deep learning ideas.

**Strengths:**

This was a good paper, I thought. The applications were fruitful, and the explanation of the theory was clear. It was well-situated in the literature, and useful and correct extensions of the method were proposed.

**Weaknesses:**

Several methods are compared, but I don't see a discussion of looking for the best available methods. Are these methods in Table 2 the best available methods? See also my questions below.

**Questions:**

1.	I guess, in my mind, it does very little good to say a method can deal with high dimensions without saying how high. There are two examples, one for 43 variables and another for 100 (genome reduced to this). Some may not consider this to be high-dimensional, so it would be better to say up front what dimension one hopes to achieve with the method (after variable reduction is done).
2.	Also, it’s important to say what density such models can attain. It’s possible with some very high-dimensional data that the models may, in fact, be very dense, a situation that can be addressed currently in the linear Gaussian or non-Gaussian case. However, this hasn’t been addressed to my knowledge for models with more general connection functions, a possible advantage of the method in this paper.
3.	There are recent papers that address a dense searches for the linear, Gaussian case and linear, non-Gaussian cases. The secret of some of these papers is to relax the Faithfulness condition, something which does allow some nonlinear functions to be addressed in a linear framework. The secret of others (like DirectLiNGAM) is to move to the linear, non-Gaussian regime. It would be really wonderful if an approach like the one in this paper could be shown to improve these algorithms (which look to be state-of-the-art) for some choices of nonlinear functions.
4.	Along these lines, I think it’s important when saying you’re outperforming methods to include in this the relaxation of assumptions that you’ve done and to compare only to other methods that relax assumptions in similar ways or that use stronger assumptions, explicitly noting this and show that with the stronger assumptions worse results are achieved. (The latter is not always the case.)
5.	The restriction to binary data for some of the theory is somewhat severe since very few real datasets consist entirely of binary variables, though perhaps I misunderstand.
6.	I’m not sure how “MNR” abbreviates “Missing At Random”; I think it must be “MAR.”


NOTE: I read the response; I thought it was convincing, so I'm raising my rating. Thanks.

---

> ### Author Response · Authors · 2023-11-18
>
> ## W1: Several methods are compared, but I don't see a discussion of looking for the best available methods. Are these methods in Table 2 the best available methods?
>
> For this paper, we mainly consider the estimation of Average Treatment Effect (ATE). For estimation of ATE, there are numerous well-adopted methods developed based on different modeling techniques. In our paper, we consider the neural-network based methods (i.e. DragonNet, X-Learner) and methods in semiparametric literature (e.g., Double Selection Estimator (DSE), Approximate Balancing Residual Estimator (ARBE), and TMLE) as baselines.
>
> Since the proposed method is based on neural networks, comparing with other neural-network based methods is essential. For semiparametric methods, DSE and ARBE are state-of-the-art methods for high-dimensional causal inference, and TMLE can be easily extended to machine learning methods in terms of nuisance parameters estimations. We summarize the performance of these methods in simulation studies and ACIC dataset in Table 1. The methods listed in Table 2 are state-of-the art methods for estimation of Conditional Average Treatment Effect (CATE), most of which are also neural network based. Although CATE estimation is not our main focus, we still listed the comparison of CATE estimation just to highlight the existing neural network-based methods on CATE estimation.

---

> ### Author Response · Authors · 2023-11-18
>
> ## Q1: It would be better to say up front what dimension one hopes to achieve with the method (after variable reduction is done)
>
> In the rebuttal, we have tried a high-dimensional example with $p=1000$ and $n=800, 1500, 3000$. Please see the results in the global response.
> The proposed method worked for all cases and outperformed the baselines.
>
> ## Q2: Also, it’s important to say what density such models can attain.
>
> We added a simulated experiment for a function with nonlinear outcome function and nonlinear treatment effect function, for which the neural network can be dense as the true functions are not exact neural network functions. Please see the results in the global response. The proposed method worked for all cases and outperformed the baselines.
>
> ## Q3: Connection with causal graph discovery
>
> This is a thoughtful question. Indeed, the sparse DNN has been used in the literature for learning causal graphs via a double regression method, see Liang and Liang (2022)[1]. The method proposed in this paper can be equally applied there.
>
> [1] Liang, S. and Liang, F. (2022) A Double Regression Method for Graphical Modeling of
> High-dimensional Nonlinear and Non-Gaussian Data. Statistics and Its Interface, in press (see also arXiv:2212.04585
>
> ## Q4:
>
> We will follow this suggestion to state the performance of the algorithm in revising the paper.
>
> ## Q5 & Q6:
>
> We are sorry for the misunderstanding. In this paper, we just assume the treatment variable is binary, while leaving the outcome and explanatory variables unrestricted. Furthermore, we have discussed the extensions of the binary treatment variable to multi-level or continuous treatment variables.

---

> > ### Comment · Reviewer_vAHH · 2023-11-20
> > **Convincing response.**
> >
> > I thought this was a convincing response, thanks.

---

> > > ### Author Response · Authors · 2023-11-20
> > > **Thank you for your feedback**
> > >
> > > Thank you very much for the constructive feedback, which we will definitely incorporate in the revision of the paper.

---

### Official Review · Reviewer_GhoU · 2023-10-31

**Soundness:** 3 good
**Presentation:** 3 good
**Contribution:** 3 good
**Rating:** 6
**Confidence:** 4

**Summary:**

This paper proposes a deep-learning-based causal inference method, that assumes the nuisance parameters are nonlinear high-dimensional models, fit by a stochastic neural network with sparsity-pursuing properties. This paper also gives theoretical guarantees, building upon a series of papers by Liang and colleagues on Bayesian neural nets.

**Strengths:**

The paper tries to address an important problem in causal inference, by taking the nuisance parameters to be high-dimensional sparse nonlinear models. State-of-the-art deep learning methods are used, e.g. stochastic/Bayesian sparsity-pursuing neural nets, to address this challenge.

The theoretical results look sound, building upon a series of earlier works on statistical guarantees of Bayesian neural nets by Liang and colleagues.

**Weaknesses:**

1. Even though I selected ``good'' in Presentation, I believe the exposition can nonetheless be significantly improved. For example, some of the key assumptions and elements are delayed to Appendix so the flow seems a bit broken, in particular in the first several sections. I strongly recommend that the authors consider revising their manuscript to make the flow smoother.

2. In the simulation, this paper considered scenarios with p=100/200 and n=10000, which do not seem to be a very high-dimensional regime. What would be the performance if we further increase the covariate dimension p or decrease the sample size n?

3. Missing references: doubly-robustness should be traced back to Robins et al. 1994 JASA. Also, Farrell et al. was cited twice in the paper (one arxiv version, one joe published version). A recent paper led by Xiaohong Chen and colleagues (Chen, Liu, Ma, Zhang, to appear in JoE) also addresses a similar problem, though their nuisance models are slightly different (Barron space). This paper should also be cited.

4. Theory-practice gap: As we know, almost all theoretical works on deep learning do not reflect practice. For instance, sparse neural nets are generally difficult to fit, as persuasively argued in Farrell, Liang, and Misra ECTA 2021. The authors are recommended to comment on this, for readers to better understand what are the key elements that allow Causal-Stonet to learn the sparse neural nets.

5. In causal inference, often times $x$ has clear scientific meaning. For sparse models, I would imagine that the input layer sparsity seems to be more important. If putting sparsity in output layer, this is essentially saying that there is some sparse nonlinear representation of the input. By assuming sparse neural nets, however, this is saying that the nonlinear representation itself is also in some sense sparse. Is it really aligned with our view of the real world or is it more like a contrived modeling assumption? I hope that the authors could further comment on the modeling philosophy adopted in this paper.

**Questions:**

1. Do the theoretical results rely on equation (6)? If so, then the theoretical results seem to go against the conventional wisdom in the deep learning literature, that is, the neural parameters themselves are not scientifically meaningful so it is not that important to learn the neural net parameters.

2. In theory, the propensity model and outcome model are both sparse models with sparsity levels lower than n^{3/16}. In linear models, the sparsity allowed is n^{1 / 2} up to log factors. Could the authors provide a heuristic explanation on the rate n^{3 / 16}?

**Details Of Ethics Concerns:**

NA.

---

> ### Author Response · Authors · 2023-11-18
>
> ## W1: I believe the exposition can nonetheless be significantly improved.
>
>
> We will follow your suggestion to move the key assumptions and elements into the main text, ensuring a smooth flow.
>
>
> ## W2: What would be the performance if we further increase the covariate dimension p or decrease the sample size n?
>
> Please refer to the experiment results in the global response, where we considered the cases with $p=1000$ and $n=800$, 1500 and 3000. The proposed method worked for all cases and outperformed the baselines.
>
> ## W3: Missing references
>
> The references will be updated as suggested: Cite Robins et al. 1994 [1] for doubly-robustness, remove the arXiv paper Farrell et al. 2018, and add Chen et al. 2024 [2].
>
> [1] Robins, J.M., Rotnitzky, A., and Zhao, L.P. (1994) Estimation of regression coefficients when some regressors are not always observed. J. Am. Stat. Assoc., 89, 846-866.
>
> [2] Chen, X., Liu, Y., Ma, S., and Zhang, Z. (2024) Causal inference of general treatment effects using neural networks with a diverging number of confounders, Journal of Econometrics, 238, 105555
>
> ## W4: Theory-practice gap
>
> Tricks for efficiently training sparse DNNs have been discussed and developed in Sun et al. [1,2]. Specifically, Sun et al. [1] recommended starting with an overparameterized DNN, while Sun et al. [2] further proposed a prior-annealing method.
> This method involves gradually incorporating the prior onto an overparameterized DNN in an annealing manner and is potentially immune to local traps. In this paper, we adopt the approach suggested by Sun et al. [1], training the sparse StoNet by initially employing an overparameterized model.
>
> [1] Y. Sun, Q. Song, and F. Liang. Consistent sparse deep learning: Theory and computation. Journal of the American Statistical Association, 117(540):1981–1995, 2022.
>
> [2] Y. Sun, W. Xiong, and F. Liang. Sparse deep learning: A new framework immune to local traps and miscalibration. NeurIPS 2021, 2021.
>
> ## W5: Modeling philosophy
>
> Thank you for your thoughtful question. We address this query from two perspectives. In biological neural networks, as discussed in [1], a wealth of empirical evidence suggests that neurons represent information using sparse distributed patterns of activity. Concerning artificial DNNs, as shown in [2] based on the random matrix theory, the stochastic gradient-based training process results in an implicit heavy-tailed self-regularization in their weights, i.e., achieving an essentially sparse representation even without a sparse penalty used during the training process.
> The proposed work makes this sparse pattern of connection weights more explicit and validates the downstream inference.
>
> [1]  S. Ahmad and J. Hawkins (2016) How do neurons operate on sparse distributed representations? A mathematical theory of sparsity, neurons and active dendrites, ArXiv:1601.00720.
>
> [2] C.H. Martin and M.W. Mahoney (2021). Implicit Self-Regularization in Deep Neural Networks: Evidence from Random Matrix Theory and Implications for Learning, J. Mach. Learn. Res., 22, 165:1-73.

---

> ### Author Response · Authors · 2023-11-18
>
> ## Q1: Identifiability of neural network parameters.
>
> Indeed, it is known that the DNN model is generally nonidentifiable due to the symmetry of the network structure. For example, the approximation of the neural networks can be invariant if one permutes the orders of certain hidden nodes, simultaneously changes the signs of certain weights and biases if tanh is used as the activation function, or re-scales certain weights and bias if Relu is used as the activation function. In this paper, following Sun et al. (2022) [1], we consider a set of networks, denoted by $\boldsymbol{\Omega}$, for which each element can be viewed as a class of equivalent DNN models. Then we define an operator $\nu(\boldsymbol{\gamma},\boldsymbol{\theta}) \in \boldsymbol{\Omega}$ that maps any neural network to $\boldsymbol{\Omega}$ via appropriate transformations such as nodes permutation, sign changes, weight rescaling. To define $\boldsymbol{\Omega}$ usually comes with constraints on the neural network, and Liang et al. (2018) [2] gives an example of such constraints. However, even if we relax this identifiability assumption on network parameters, the validity of StoNet as an approximator to DNN can still be justified, since StoNet and DNN have asymptoticlaly equivalent loss functions (see equation (5)).
>
> [1] Y. Sun, Q. Song, and F. Liang. Consistent sparse deep learning: Theory and computation.
> Journal of the American Statistical Association, 117(540):1981–1995, 2022.
>
> [2] Liang, F., Li, Q., and Zhou, L. (2018), “Bayesian neural networks for selection of drug sensitive genes,” Journal of the American Statistical Association, 113, 955–972.
>
> ## Q2: Could the authors provide a heuristic explanation on the rate $O(n^{3/16})$?
>
> Again, this is a thoughtful question. A heuristic explanation is that the size of the true DNN model is allowed to increase with the sample size $n$ at the rate $O(n^{3/16})$, while ensuring the validity of inference for the treatment effect.
>
> We note that this rate is derived based on the Laplace approximation error
> for high-dimensional Bayesian sparse nonlinear models, as presented in Theorem 2.3 by Sun et al. 2022 [1].
> Specifically, Sun et al. (2023) showed that the Laplace approximation error is of $O(r_n^4/n)$,
> a general result for the MAP-based posterior mean approximation in Bayesian sparse nonlinear models.
>
> We believe that this rate can be improved
> under the frequentist framework, but the proof appears to be challenging. Refer to our discussions about Farrell et al. (2021) [2] in the  ``Related Works'' paragraph for further insights.
>
> [1] Sun, Q. Song, and F. Liang. Consistent sparse deep learning: Theory and computation.
> Journal of the American Statistical Association, 117(540):1981–1995, 2022.
>
> [2] M. Farrell, Tengyuan Liang, and S. Misra. Deep neural networks for estimation and inference.
> Econometrica, 89:181–213, 2021.

---

> ### Comment · Reviewer_GhoU · 2023-11-20
> **thank you for your response**
>
> Thank you for the authors' thoughtful response. I am satisfied with the response. If there were a score 7, I would have been willing to improve the current score (6) to 7.

---

> > ### Author Response · Authors · 2023-11-20
> > **Thank you for your feedback**
> >
> > Thank you very much for your encouraging feedback and your kindness in increasing the score.

---

### Official Review · Reviewer_7o25 · 2023-11-01

**Soundness:** 3 good
**Presentation:** 2 fair
**Contribution:** 2 fair
**Rating:** 6
**Confidence:** 3

**Summary:**

This paper introduces an approach to causal inference tailored for high-dimensional complex data. This approach draws upon recent developments in deep learning techniques, including sparse deep learning and stochastic neural networks. By leveraging these techniques, the proposed approach effectively tackles both the high dimensionality and the complexity of the underlying data generation process. Moreover, it can handle scenarios with missing values in the datasets.

**Strengths:**

- The proposed method Causal-StoNet is proven to have the universal approximation ability, making it a versatile tool for modeling outcome and propensity score functions.

- The paper provides a strong theoretical foundation for its approach, including proofs and mathematical support, enhancing its reliability.

**Weaknesses:**

1/ It seems to me that the proposed method is a direct application of stochastic neural networks. Please clearly highlight technical innovation of the proposed method.

2/ Can the authors explain which component of their method make it works well for high-dimensional confounder X? Why the existing methods such as: CEVAE, CFR Net are not possible to deal with high dimensions of X?

3/ It seems to me that many concepts and notations in the paper are unexplained. For example, why do we choose $\sigma_{0,n}^n$ to be a very small number while $\sigma_{1,n}^2$ is relatively large? Is there any rationale for this? What is $\pi(\theta)$ in Eq.~9? Is it the prior?

4/ Since $Y_{mis}^i$ is unobserved, how do you minimise Eq.~9?

**Questions:**

Please see section weaknesses

---

> ### Author Response · Authors · 2023-11-18
>
> ## W1: It seems to me that the proposed method is a direct application of stochastic neural networks. Please clearly highlight technical innovation of the proposed method.
>
> The technical innovations of this paper lie in two  aspects. First, we propose using the stochastic neural network as a general tool for modeling natural systems by incorporating middle-level observations as visible units in some hidden layers. While we acknowledge that the stochastic neural network has been proposed in the literature, our utilization of it is innovative. This approach enables the modeling of many complex systems by deep neural networks (DNNs) in a natural way, facilitating downstream inference.
>
> Second, to ensure that Causal-StoNet can handle high-dimensional data and that the resulting statistical inference is valid, we impose a mixture Gaussian prior on each connection of the Causal-StoNet (see Equation (8)). Under some regularity conditions, we establish consistency for structure selection, outcome function estimation, and propensity score function estimation. Consequently, we establish the validity of statistical inference for the treatment effect. Without the sparsity prior, statistical inference with Causal-StoNet cannot be valid.
>
> Here, we would like to emphasize that for many neural network-based methods, justifying the validity of downstream inference can be challenging unless the dataset size is sufficiently large and the network size is relatively small. On the other hand, we are aware that deep neural networks trained with stochastic gradient-based methods possess the property of implicit heavy-tailed self-regularization (see Ref. [1]). However, since the self-regularization phenomenon cannot be rigorously measured, at least for now, establishing the validity of downstream inference can still be challenging. Our method provides a theoretical guarantee for the validity of the neural network-based causal inference.
>
> [1] C.H. Martin and M.W. Mahoney (2021). Implicit Self-Regularization in Deep Neural Networks: Evidence from Random Matrix Theory and Implications for Learning,
> J. Mach. Learn. Res., 22, 165:1-73.
>
> ## W2: Can the authors explain which component of their method make it works well for high-dimensional confounder X? Why the existing methods such as: CEVAE, CFR Net are not possible to deal with high dimensions of X?
>
> As mentioned above, the sparsity prior enables Causal-StoNet to perform effectively with high-dimensional confounders. Moreover, the resulting sparse structure ensures consistency of the estimation and the validity of the subsequent causal inference.
>
> In contrast, both CEVAE and CFRNet utilize fully-connected neural networks, where the consistency of estimation cannot be ensured. Consequently, validating the subsequent causal inference becomes challenging. For comparison, we have applied CEVAE and CFRNet to the ACIC example, see Table 2. The comparison indicates that both methods are inferior to the proposed one for the example.
>
> References: Shalit, Uri, Fredrik D. Johansson, and David Sontag. “Estimating individual treatment effect: generalization bounds and algorithms." International Conference on Machine Learning. PMLR, 2017.
>
> Johansson, Fredrik, Uri Shalit, and David Sontag. “Learning representations for counterfactual inference.” International conference on machine learning. PMLR, 2016.
>
> ## W3: It seems to me that many concepts and notations in the paper are unexplained. For example, why do we choose $\sigma_{0, n}^n$ to be a very small number while $\sigma_{1, n}^n$ is relatively large? Is there any rationale for this? What is $\pi(\theta)$ in Eq.~9? Is it the prior?
>
> We will double-check and clarify the notations in the revision. The restriction on the noise is needed to ensure that the Causal StoNet can have asymptotically equivalent loss function as DNN (see equation (5)), which is crucial in justifying that the Causal-SoNet is a good approximator of the underlying DNN and hence a valid universal learner for the treatment  model and the outcome model. The $\pi(\theta)$ in equation (9) refers to the mixture Gaussian prior.
>
> ## W4: Since is $\boldsymbol{Y_{\text{miss}}}^i$ unobserved, how do you minimise Eq.~9?
>
> Equation (9) can be minimized by solving a mean-field equation, as given in (10), using an adaptive SGMCMC algorithm.
> The Adaptive SGMCMC algorithm operates by iterating over the following two steps:
>
> (1) Sampling $\boldsymbol{Y_{\text{miss}}}^i$ from $\pi(\boldsymbol{Y_{\text{miss}}}^i | \boldsymbol{X}, A, \boldsymbol{Y}, \boldsymbol{\theta})$;
>
> (2) update $\boldsymbol{\theta}$ based on the sampled latent variables.
>
> For more details of the algorithm, please refer to section 3.2. Additionally, Appendix A1 provides the pseudo-code for the training algorithm of Causal-StoNet, and Appendix A6 includes a brief discussion of the convergence analysis of the algorithm.

---

> > ### Comment · Reviewer_7o25 · 2023-11-20
> > **Response to authors**
> >
> > Many thanks to the authors for their clarifications. Integrating these insights into the paper would significantly enhance its clarity. Additionally, I appreciate the authors' inclusion of additional experimental results. With all these improvements taken into account, I would like to elevate the score.

---

> > > ### Author Response · Authors · 2023-11-20
> > > **Thank you for your feedback**
> > >
> > > Thank you very much for kindly raising the score. We will certainly follow your suggestions to clarify any confusing concepts or notations during the revision.

---

### Author Response · Authors · 2023-11-18
**Simulated Experiment with p = 1000, n = 800, 1500, 3000.**

We have run additional simulated experiments for $p = 1000$, and $n = 800, 1500, 3000$. Here, $p$ denotes the dimension of the covariates.

## Dataset Generation

10 simulation datasets are generated. For most procedures of simulation dataset generation, we follow Lei and Candes (2021) [1]:

1. Generate $e, z_1, \cdots, z_{100}$ independently from a truncated standard normal distribution on the interval $[-10, 10]$. Set $x_i = \frac{e+z_i}{\sqrt{2}}$ for $i = 1, \cdots, 100$, making the covariates highly correlated.

2. The propensity score $e(\boldsymbol{x}) = \frac{1}{4}(1+\beta_{2,4}(\Phi(x_1)))$, where $\beta_{2,4}$ is the cdf of the beta distribution with shape parameters (2, 4), and $\Phi$ denotes the cdf of the standard normal distribution.
This ensures that $e(x) \in [0.25, 0.5]$, thereby providing sufficient overlap. Treatment $A_i$ is hence generated by a Bernoulli distibution with the probablity of success being $e(x_i)$,
and resampling from the treatment and control groups has been performed for ensuring that the dataset contains equal numbers of treatment and control samples.

3. For simulation of observed outcome, we consider

    \begin{equation}
            y_i = c(\boldsymbol{x_i})+ \tau A_i+\eta_i*A_i + \sigma z_i, \quad i=1,2,\ldots,n
    \end{equation}

    \begin{equation}
                    c(\boldsymbol{x_i}) =5 x_{2}/(1+x_{1}^2) + 2 x_{5}
    \end{equation}

    where $\eta(x_i)=f(x_{1})f(x_{2}) - E(f(x_{1})f(x_{2}))$ and $f(x)=\frac{2}{1+exp(-12(x-0.5))}$. In other words,  we set treatment effect $\tau(x_i)=\tau +\eta_i$. We generated the data under the setting $\tau=3$ and $\sigma=0.25$ with $n_{train} \in \\{800, 1500, 3000\\}$, $n_{val} \in \\{160, 300, 600\\}$, $n_{test} \in \\{160, 300, 600\\}$.

## Baselines

For this experiment, since both the outcome and treatment effect functions are nonlinear, we consider TMLE (ensemble), Dragonnet, and X-Learner as baselines for ensuring a fair comparison. For TMLE (ensemble), we use the ensemble of lasso and XGBoost to estimate the nuisance functions. Both Dragonnet and X-learner are neural network-based models.

## Results

The following tables show the mean absolute error (MAE) of ATE estimation across 10 simulation datasets, with numbers in the parentheses being the standard error of the MAE.

**MAE of Simulation with n=800, p=1000**
|               | Causal-StoNet   | TMLE(ensemble)  | X-Learner       | Dragonnet       |
|---------------|-----------------|-----------------|-----------------|-----------------|
| In-sample     | 0.1642 (0.0468) | 0.4552 (0.0387) | 0.6155 (0.0935) | 0.4191 (0.0569) |
| Out-of-sample | 0.2079 (0.0654) | 0.5068 (0.0473) | 0.5754 (0.0946) | 0.4740 (0.0599) |

 **MAE of Simulation with n=1500, p=1000**
|               | Causal-StoNet   | TMLE(ensemble)  | X-Learner       | Dragonnet       |
|---------------|-----------------|-----------------|-----------------|-----------------|
| In-sample     | 0.0853 (0.0265)| 0.2784 (0.0165)| 0.5771 (0.1002) | 0.2946 (0.0789)|
| Out-of-sample | 0.1581 (0.0246) | 0.3938 (0.0218)| 0.5667 (0.1058)| 0.3263 (0.0836)|

 **MAE of Simulation with n=3000, p=1000**
|               | Causal-StoNet   | TMLE(ensemble)  | X-Learner       | Dragonnet       |
|---------------|-----------------|-----------------|-----------------|-----------------|
| In-sample     | 0.0611 (0.0202) | 0.1511 (0.0124) | 0.4179 (0.0800) | 0.1983 (0.0454) |
| Out-of-sample | 0.0870 (0.0336) | 0.1704 (0.0125) | 0.4291 (0.0871) | 0.2227 (0.0471) |


Reference:  [1] Lihua Lei and Emmanuel J. Cand\'es. Conformal Inference of Counterfactuals and Individual Treatment Effects. Journal of the Royal Statistical Society Series B: Statistical Methodology, 83(5):911–938, 10 2021. ISSN 1369-7412.

---

### Meta-Review · Area_Chair_kyAQ · 2023-12-08

**Metareview:**

This paper provides a new perspective on causal inference with deep neural networks tailored towards high-dimensional models. All reviewers are in favor of acceptance.

**Justification For Why Not Higher Score:**

Although all reviewers are in support of acceptance, the paper has some minor flaws and could be improved.

**Justification For Why Not Lower Score:**

Uniform support from reviewers.

---

### Decision · Program_Chairs · 2024-01-16

Accept (poster)